

# Density matrices in integrable face models

**Holger Frahm**[⋆] **and Daniel Westerfeld**[†]

Institut für Theoretische Physik, Leibniz Universität Hannover
Appelstraße 2, 30167 Hannover, Germany

⋆ frahm@itp.uni-hannover.de, † daniel.westerfeld@itp.uni-hannover.de

## Abstract

Using the properties of the local Boltzmann weights of integrable interaction-round-a-face (IRF or face) models we express local operators in terms of generalized transfer matrices. This allows for the derivation of discrete functional equations for the reduced density matrices in inhomogeneous generalizations of these models. We apply these equations to study the density matrices for IRF models of various solid-on-solid type and quantum chains of non-Abelian $su(2)_3$ or Fibonacci anyons. Similar as in the six vertex model we find that reduced density matrices for a sequence of consecutive sites can be 'factorized', i.e. expressed in terms of nearest-neighbour correlators with coefficients which are independent of the model parameters. Explicit expressions are provided for correlation functions on up to three neighbouring sites.

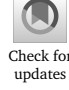

# 1  Introduction

Exact Bethe ansatz solutions of integrable lattice models provide valuable insights into properties which can be related to their spectrum such as thermodynamic properties and the nature of low energy excitations. The computation of general correlation functions in this framework is much more involved. For certain integrable vertex models and in particular the spin-1/2 Heisenberg model, however, manageable expressions for correlation functions have been obtained using methods based on the representation theory of quantum algebras, functional equations of $q$-Knizhnik-Zamolodchikov (qKZ) type, or the algebraic Bethe ansatz [1–5].

Considering inhomogeneous generalizations of these models a remarkable property of the corresponding reduced density matrices has been established: in Refs. [6–8] it was found that correlation functions of spins on $N$ consecutive sites in the ground state of the infinite length antiferromagnetic Heisenberg chain are solutions of the qKZ equation (or a reduced version thereof) and can be expressed as sums of terms factorizing into products of nearest neighbour (two-point) functions of the generalized models. Their coefficients are recursively defined elementary functions of $N$ spectral parameters and do not depend on model parameters such as the system size or choice of inhomogeneities. While this approach based on the qKZ equations is limited to infinite chains the factorization was also observed in studies of the three-site reduced density matrix for isotropic chains of finite length (or at finite temperature) [9, 10]. Later, using the fermionic structure in the space of operators of the XXZ model [11, 12], this property has been proven to hold for arbitrary correlation functions of the XXZ Heisenberg chain in an external magnetic field and at finite temperature [13].

In another approach to the computation of correlation functions of the Heisenberg chain discrete functional equations of reduced qKZ-type have been derived starting from the local properties of the six vertex model [14, 15]. These equations can be shown to characterize the generalized $N$-site reduced density matrix $D_N$ of the model uniquely when complemented with an asymptotic reduction relating the latter to $D_{N-1}$ when one of the spectral parameters is sent to infinity [15]. Taking the factorized form of the reduced density matrix from Ref. [7] as an ansatz it is found that the latter does indeed satisfy these relations. Therefore, this approach provides an alternative, though not constructive proof of the factorization property.

Less is known for integrable interaction-round-a-face (IRF) models. For an important class of IRF models, the Andrews-Baxter-Forrester (ABF) series of solid-on-solid (SOS) models, Baxter's corner transfer matrix [16] has been used to compute local height probabilities in the infinite lattice [17]. Vertex operators for the ABF models introduced in this approach can be related to representations of quantum group symmetries present in the infinite system and their correlation functions satisfy qKZ equations [18] and the algebra formed by the vertex operators has been bosonized to obtain integral representations for multi-point local height probabilities in restricted SOS (or RSOS) models in the thermodynamic limit [19]. As for the vertex models the integrability of IRF models is based on a Yang-Baxter equation satisfied by the Boltzmann weights. For the SOS models these weights can be arranged in an $R$-matrix depending on an additional dynamical parameter. Due to this formulation certain aspects of the Quantum Inverse Scattering Method can be employed for the analysis of these models: the corresponding modified (dynamical) Yang-Baxter algebra allows for the solution of the spec-

tral problem by means of an algebraic Bethe ansatz [20] or an adaption [21] of the framework of Sklyanin's Separation of Variables [22]. Similarly, the inverse problem relating local spin operators to elements of the Yang-Baxter algebra [23, 24] has been solved for the dynamical vertex models and also allows to express local heights in this context [25–27]. A remaining difficulty for the application of the algebraic Bethe ansatz approach to the calculation of local height probabilities arises from the expressions of matrix elements of local height operators. Even for the simple case of the cyclic SOS model with rational crossing parameter these appear to be more complicated than in the (non-dynamical) six-vertex model [26, 27]. Factorization properties of the reduced density matrices in integrable face models have, to our knowledge, not been studied so far.

In this paper we address some of the issues appearing in the calculation of correlation functions in generic face models in particular on finite lattices without refering to a possible formulation as a dynamical vertex model. In the following section we specify the possible configurations of an IRF model and the Hilbert space of the related quantum chains of non-Abelian anyons. The Boltzmann weights of local configurations for these models are used to define a family of transfer matrices with generalized boundary conditions. In Section 3 the solution of the inverse problem is presented for a class of local operators for an inhomogeneous IRF model with certain local properties of the Boltzmann weights, in particular unitarity, together with an initial condition for their dependence on the spectral parameter. Expectation values of these operators in eigenstates of the transfer matrix are encoded in generalized $N$-point density matrices $D_N(\lambda_1, \ldots, \lambda_N)$ with independently chosen spectral parameters $\lambda_n$. In Section 4 we derive a set of linear functional equations of reduced qKZ-type for $D_N$ which holds for a discrete set of values of the spectral parameter $\lambda_N$.

In a final section we consider several SOS models and the related chain of Fibonacci anyons where we find that these functional equations together with the analytical properties of the density matrices derived from those of the local Boltzmann weights do in fact uniquely determine the $D_N$ in these models. Assuming that the factorization property for the $N$-point density matrices mentioned above also holds for integrable IRF models we propose an algorithm for the efficient computation of the structure coefficients.

## 2 Integrable face models

Interaction-round-a-face (IRF or face) models are classical statistical models defined on a square lattice with a spin (or height) $a_\ell$ assigned to each site $\ell$. The heights take values from a set $\mathfrak{S}$ possibly subject to adjacency rules constraining their values on neighbouring vertices. These rules are conveniently presented in the form of a graph with nodes $a \in \mathfrak{S}$ and adjacency matrix

$$A_{ab} \equiv \begin{cases} 1, & \text{spins } a \text{ and } b \text{ are allowed to be adjacent} \\ 0, & \text{spins } a \text{ and } b \text{ are not allowed to be adjacent} \end{cases}. \tag{1}$$

The energy of the face model for a given height configuration is determined by local Boltzmann weights depending on the spins on the vertices surrounding an elementary face [17]. These weights are allowed to depend on an arbitrary (spectral) parameter $u$ and are depicted graphically as

$$W\begin{pmatrix} d & c \\ a & b \end{pmatrix} u = \boxed{\begin{matrix} d & & c \\ & u & \\ a & & b \end{matrix}} = a \left\langle \begin{matrix} d \\ u \\ b \end{matrix} \right\rangle c . \tag{2}$$

Related quantum models describing interacting non-Abelian anyons in one spatial dimension can be obtained from face models in their Hamiltonian limit, see e.g. [28–31]. Mathematically these anyon models can be described by braided tensor categories [32] consisting of a collection of objects $\{\psi_a\}$ (including an identity). They are equipped with a set of fusion rules

$$\psi_a \otimes \psi_b = \bigoplus_c N^c_{ab} \psi_c \,. \tag{3}$$

This rule for the fusion of objects $\psi_a$ and $\psi_b$ into $\psi_c$ can be represented graphically, where the vertex (to be read from top-left to bottom-right)

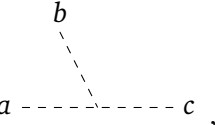

is allowed provided that $N^c_{ab} \neq 0$.

The fusion rules allow to construct the Hilbert space of a chain of $L$ interacting anyons with topological charge $\psi_{a_*}$ and their possible local interactions. In this paper we consider tensor categories which are free of multiplicities, i.e. $N^c_{ab} \in \{0,1\}$, and starting with an auxiliary anyon $\psi_{a_0}$ an orthogonal basis of 'fusion path' states is constructed by fusing $\psi_{a_\ell}$ and $\psi_{a_*}$ into $\psi_{a_{\ell+1}}$ resulting in

$$|a_0 a_1 \dots a_L\rangle \text{ with } N^{a_{\ell+1}}_{a_\ell a_*} = 1 \text{ for } \ell = 0 \dots L-1 \tag{4}$$

or, using the graphical representation of these consecutive fusion processes,

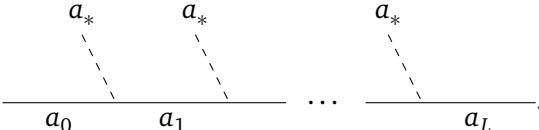

Note that the sequence $\{a\} = (a_0 a_1 \dots a_L)$ coincides with a possible height configuration along a horizontal line of vertices in the face model on a lattice with $L \times N$ faces provided that the possible topological charges are labelled by the elements of $\mathfrak{S}$ and $N^{a_{\ell+1}}_{a_\ell a_*} = A_{a_\ell a_{\ell+1}}$.

Clearly the Hilbert space $\mathcal{H}^L$ spanned by these fusion paths can be decomposed into sectors $\mathcal{H}^L_{\alpha\beta}$ labeled by the auxiliary spins $\alpha = a_0$ and $\beta = a_L$. Below we consider anyon chains (face models) with periodic boundary conditions (in the horizontal direction). Therefore we have to identify $a_0$ and $a_L$ which allows to remove the label $a_0$ from the basis states (4). As a result the model is defined on the Hilbert space ('quantum space')

$$\mathcal{H}^L_{\text{per}} = \bigoplus_{\alpha \in \mathfrak{S}} \mathcal{H}^L_{\alpha\alpha} \,. \tag{5}$$

Note that states in $\mathcal{H}^L_{\text{per}}$ correspond to periodic paths of length $L$ on the adjacency graph.

Similarly, the sequences $\{\underline{\alpha}\} = (\alpha_0 \alpha_1 \dots \alpha_N)$ with $N^{\alpha_{n+1}}_{\alpha_n a_*} = A_{\alpha_n \alpha_{n+1}} = 1$ of heights on vertical lines can be identified with fusion paths spanning an 'auxiliary' Hilbert space $\mathcal{V}^N$ of anyons. Here and below we use greek indices for the auxiliary height variables on vertical lines and latin indices to label the height variables corresponding to an anyonic quantum state on horizontal lines.

We represent the matrix elements of generic linear operators $B$ on the space $\mathcal{V}^N \hat{\otimes} \mathcal{H}^L$ as [1]

$$\left( \langle \boldsymbol{a} | \hat{\otimes} \langle \underline{\boldsymbol{\alpha}} | \right) B \left( | \underline{\boldsymbol{\beta}} \rangle \hat{\otimes} | \boldsymbol{b} \rangle \right) =$$

$$
\begin{array}{c}
\alpha_0 = a_0 \qquad \cdots \qquad \beta_0 = a_L \\
\alpha_1 \qquad\qquad\qquad\qquad \beta_1 \\
\vdots \qquad\qquad\quad B \qquad\qquad\quad \vdots \\
\alpha_{N-1} \qquad\qquad\qquad\qquad \beta_{N-1} \\
\alpha_N = b_0 \qquad \cdots \qquad \beta_N = b_L
\end{array}
\tag{6}
$$

The matrix elements of $B$ in $\mathcal{V}^N$ are linear operators on $\mathcal{H}^L$ (and vice versa):

$$B^{\{\underline{\alpha}\}\{\underline{\beta}\}} = \langle \underline{\boldsymbol{\alpha}} | B | \underline{\boldsymbol{\beta}} \rangle, \quad B^{\{a\}}_{\{b\}} = \langle \boldsymbol{a} | B | \boldsymbol{b} \rangle. \tag{7}$$

As an example, we define an operator $T(u)$ on $\mathcal{V}^1 \hat{\otimes} \mathcal{H}^L$ from a single row of the the Boltzmann weights (2):

$$\left( \langle \boldsymbol{a} | \hat{\otimes} \langle \underline{\boldsymbol{\alpha}} | \right) T(u) \left( | \underline{\boldsymbol{\beta}} \rangle \hat{\otimes} | \boldsymbol{b} \rangle \right) = \langle \boldsymbol{a} | T^{\alpha_0 \beta_0}_{\alpha_1 \beta_1}(u) | \boldsymbol{b} \rangle$$

$$
= \quad
\begin{array}{c}
\alpha_0 = a_0 \quad a_1 \qquad\quad a_{L-1} \quad a_L = \beta_0 \\
\boxed{u - u_1 \quad \Big| \quad \cdots \quad \Big| \quad u - u_L} \\
\alpha_1 = b_0 \quad b_1 \qquad\quad b_{L-1} \quad b_L = \beta_1
\end{array}
\tag{8}
$$

$$= \prod_{i=1}^{L} W \left( \begin{array}{cc} a_{i-1} & a_i \\ b_{i-1} & b_i \end{array} \middle| u - u_i \right) \delta_{a_0, \alpha_0} \delta_{a_L, \beta_0} \delta_{b_0, \alpha_1} \delta_{b_L, \beta_1}.$$

Here, the complex numbers $\{u_i\}_{i=1}^L$ parameterize local variations in the interactions around a face. Taking the trace in $\mathcal{V}^1$ this gives the transfer matrix of the inhomogeneous face model with periodic boundary conditions in the horizontal direction

$$t(u) = \mathrm{tr}_{\mathcal{V}^1} T(u) = \sum_{\alpha, \beta} T^{\alpha \alpha}_{\beta \beta}(u),$$

$$
\langle \boldsymbol{a} | t(u) | \boldsymbol{b} \rangle = \quad
\begin{array}{c}
a_0 \qquad\quad a_1 \qquad\quad a_{L-1} \quad a_0 = a_L \\
\boxed{u - u_1 \quad \Big| \quad \cdots \quad \Big| \quad u - u_L} \\
b_0 \qquad\quad b_1 \qquad\quad b_{L-1} \quad b_0 = b_L
\end{array}
\tag{9}
$$

For later use we also define the following generalized transfer operators

$$T_{\alpha \beta}(u) = \sum_{\gamma \delta} T^{\gamma \delta}_{\alpha \beta}(u), \qquad T^{\alpha \beta}(u) = \sum_{\gamma \delta} T^{\alpha \beta}_{\gamma \delta}(u), \tag{10}$$

mapping $\mathcal{H}^L_{\alpha \beta} \to \mathcal{H}^L$ and vice versa. Note that $T_{\alpha \beta}(u) T^{\alpha \beta}(v)$ is a linear operator on $\mathcal{H}^L$ which leaves $\mathcal{H}^L_{\mathrm{per}}$ invariant.

---

[1]The symbol $\hat{\otimes}$ indicates that the index of the joint vertex of the two factors coincides.

Degenerations of this construction are the elementary operators $(E^\alpha_\beta)_n$ and $E^{\alpha_{n_1}\dots\alpha_{n_2}}_{\beta_{n_1}\dots\beta_{n_2}}$ on $\mathcal{H}^L$

$$\langle \boldsymbol{a} | \left( E^\alpha_\beta \right)_n | \boldsymbol{b} \rangle = \delta_{a_n,\alpha}\, \delta_{b_n,\beta} \prod_{j\neq n} \delta_{a_j b_j}$$

$$= \underset{\substack{a_{n-1} \quad\quad\quad\quad a_{n+1} \\[2pt] \beta = b_n}}{\overset{\alpha = a_n}{\diamondsuit}}$$

$$\langle \boldsymbol{a} | E^{\alpha_{n_1}\dots\alpha_{n_2}}_{\beta_{n_1}\dots\beta_{n_2}} | \boldsymbol{b} \rangle = \prod_{k=n_1}^{n_2} \delta_{a_k,\alpha_k}\, \delta_{b_k,\beta_k} \prod_{j\notin\{n_1\dots n_2\}} \delta_{a_j b_j}$$

$$= \underset{\substack{a_{n_1-1} \quad\quad\quad\quad\quad\quad a_{n_2+1} \\[2pt] \beta_{n_1}=b_{n_1} \quad \cdots \quad \beta_{n_2}=b_{n_2}}}{\overset{\alpha_{n_1}=a_{n_1} \quad \dots \quad \alpha_{n_2}=a_{n_2}}{\hexagon}}$$

(11)

acting locally on a single site $n$ or on sequences of $n_2 - n_1 + 1$ of neighbouring sites (subject to constraints for the heights on the neighbouring sites as a consequence of the adjacency conditions).

A face model is said to be integrable if its transfer matrix commutes for different values of the spectral parameter, i.e. $[t(u), t(v)] = 0$. This is guaranteed by a local condition on the Boltzmann weights: the face version of the Yang-Baxter equation reads

$$\sum_{g\in\mathfrak{S}} W \begin{pmatrix} f & g \\ a & b \end{pmatrix} u-v \Big) W \begin{pmatrix} f & e \\ g & d \end{pmatrix} v \Big) W \begin{pmatrix} g & d \\ b & c \end{pmatrix} u \Big)$$

$$= \sum_{g\in\mathfrak{S}} W \begin{pmatrix} f & e \\ a & g \end{pmatrix} u \Big) W \begin{pmatrix} a & g \\ b & c \end{pmatrix} v \Big) W \begin{pmatrix} e & d \\ g & c \end{pmatrix} u-v \Big),$$

(12)

or, in the graphical notation,

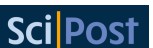

(13)

where heights on nodes with a solid circle are summed over and heights connected by dotted lines are taken to be equal.

We also assume a unitarity condition

$$\sum_{e\in\mathfrak{S}} W \begin{pmatrix} d & e \\ a & b \end{pmatrix} u \Big) W \begin{pmatrix} d & c \\ e & b \end{pmatrix} -u \Big) = \rho(u)\rho(-u)\delta_{ac},$$

(14)

crossing symmetry

$$W \begin{pmatrix} d & c \\ a & b \end{pmatrix} u \Big) = W \begin{pmatrix} c & b \\ d & a \end{pmatrix} \lambda-u \Big),$$

(15)

and the initial condition

$$W \begin{pmatrix} d & c \\ a & b \end{pmatrix} 0 = \boxed{\begin{array}{c} d \quad\quad c \\ 0 \\ a \quad\quad b \end{array}} = \delta_{a,c}. \tag{16}$$

Here $\lambda$ is the crossing parameter and $\rho(u)$ a function, both are model-dependent. We assume $\rho(0)^2 = 1$ which can always be reached by rescaling the Boltzmann weights.

## 3 Reduced density matrices

A complete characterization of the models introduced above requires the computation of generic correlation functions, which can in turn be expressed through reduced density matrices, i.e. matrix elements of the operators (11)

$$\frac{1}{\langle \Phi_0 | \Phi_0 \rangle} \langle \Phi_0 | E^{\alpha_{n_1} \dots \alpha_{n_2}}_{\beta_{n_1} \dots \beta_{n_2}} | \Phi_0 \rangle. \tag{17}$$

Here $|\Phi_0\rangle \in \mathcal{H}^L_{\text{per}}$ is the ground state of the model. More generally one may consider the right (left) eigenvectors $|\Phi\rangle$ ($\langle\Phi|$) of the transfer matrix corresponding to a particular eigenvalue $\Lambda(u)$, i.e. $t(u)|\Phi\rangle = \Lambda(u)|\Phi\rangle$ ($\langle\Phi|t(u) = \langle\Phi|\Lambda(u)$).

An important step towards the calculation of these quantities in integrable (vertex) models has been the solution of the 'inverse problem', i.e. expressing local operators through elements of the Yang-Baxter algebra. This has been achieved for the six-vertex and related models in [23, 24, 33]. In Refs. [25, 26] this construction has been generalized to local spin and local height operators in face models allowing for a formulation as a dynamical vertex model [20, 34, 35]. Here we formulate the solution to the inverse problem for a general integrable face model, i.e. without using the existence of an $R$-matrix satisfying a dynamical Yang-Baxter equation, by expressing the elementary height operators (11) in terms of the single-row operators (8), (10) introduced above:

**Theorem 1.** *The local operator* $\left( E^{\alpha}_{\beta} \right)_n$, $1 \leq n < L$, *can be expressed as*

$$\left( E^{\alpha}_{\beta} \right)_n = \prod_{k,\ell=1}^{L} \frac{1}{\rho(u_k - u_\ell)} \left( \prod_{k=1}^{n-1} t(u_k) \right) T_{\alpha\beta}(u_n) T^{\alpha\beta}(u_{n+1}) \left( \prod_{k=n+2}^{L} t(u_k) \right). \tag{18}$$

*We use the convention that empty products are 1.*

*Proof.* Let us first proof the statement for an easy example where the chain length is $L = 2$, i.e. two faces per row. For $|a\rangle, |b\rangle \in \mathcal{H}^2_{\text{per}}$ we calculate:

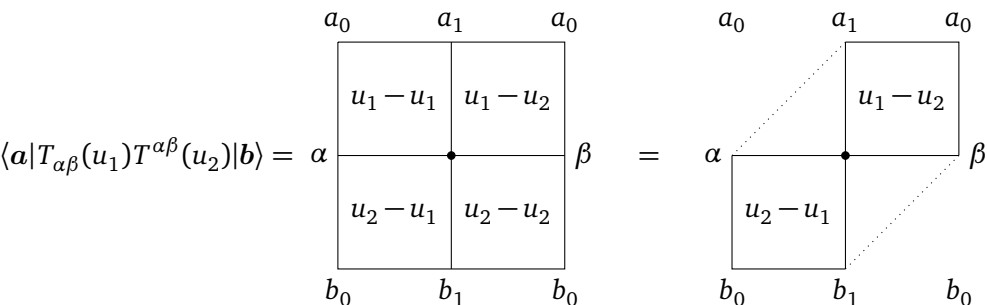

$$= \prod_{k,\ell=1}^{2} \rho(u_k - u_\ell)\delta_{a_0 b_0}\delta_{a_1\alpha}\delta_{b_1\beta} = \prod_{k,\ell=1}^{2} \rho(u_k - u_\ell)\langle a| \left(E_\beta^\alpha\right)_1 |b\rangle.$$

For general $L$ the procedure works similarly, see Appendix A. $\qquad\square$

Unitarity of the Boltzmann weights implies

$$\prod_{\ell=1}^{L} t(u_\ell) = \prod_{k,\ell=1}^{L} \rho(u_k - u_\ell)\mathbb{1}. \tag{19}$$

This allows to reformulate the theorem as:

$$\left(E_\beta^\alpha\right)_n = \left(\prod_{k=1}^{n-1} t(u_k)\right) T_{\alpha\beta}(u_n)T^{\alpha\beta}(u_{n+1}) \left(\prod_{k=1}^{n+1} t^{-1}(u_k)\right). \tag{20}$$

Note that the operators $\left(E_\beta^\alpha\right)_L$ can only be represented in the form (18) for $\alpha = \beta$.

Similarly an elementary operator acting on sequences of neighbouring sites can be expressed as ($1 \le n_1 \le n_2 < L$)

$$
\begin{aligned}
E_{\beta_{n_1}\ldots\beta_{n_2}}^{\alpha_{n_1}\ldots\alpha_{n_2}} = \prod_{k,\ell=1}^{L} \frac{1}{\rho(u_k - u_\ell)} &\left(\prod_{k=1}^{n_1-1} t(u_k)\right) \times \\
&\times T_{\alpha_{n_1}\beta_{n_1}}(u_{n_1}) \left(\prod_{k=n_1+1}^{n_2} T_{\alpha_k\beta_k}^{\alpha_{k-1}\beta_{k-1}}(u_k)\right) T^{\alpha_{n_2}\beta_{n_2}}(u_{n_2+1}) \left(\prod_{k=n_2+2}^{L} t(u_k)\right) \\
= &\left(\prod_{k=1}^{n_1-1} t(u_k)\right) T_{\alpha_{n_1}\beta_{n_1}}(u_{n_1}) \times \\
&\times \left(\prod_{k=n_1+1}^{n_2} T_{\alpha_k\beta_k}^{\alpha_{k-1}\beta_{k-1}}(u_k)\right) T^{\alpha_{n_2}\beta_{n_2}}(u_{n_2+1}) \left(\prod_{k=1}^{n_2+1} t^{-1}(u_k)\right).
\end{aligned}
\tag{21}
$$

We emphasize that this construction and proof is valid for all face models whose Boltzmann weights satisfy the unitarity conditions.

Let us now introduce the operators $\tilde{T}_N^{\{\alpha\}\{\beta\}} : \mathcal{H}^L \to \mathcal{H}^L$, corresponding to $N = n_2 - n_1 + 2$ consecutive rows of the face model with independent spectral parameters $\lambda_{n_1}, \ldots, \lambda_{n_2+1}$ and fixed sequences $\alpha = (\alpha_\ell)_{\ell=n_1-1}^{n_2+1}$ and $\beta = (\beta_\ell)_{\ell=n_1-1}^{n_2+1}$ of auxiliary indices (although not written explicitly the index $N$ is to be understood as the combination $(n_1, n_2 = n_1 + N - 2)$)

$$\tilde{T}_N(\lambda_{n_1}, \ldots, \lambda_{n_2+1})^{\{\alpha\}\{\beta\}} = \prod_{k=n_1}^{n_2+1} T_{\alpha_k\beta_k}^{\alpha_{k-1}\beta_{k-1}}(\lambda_k). \tag{22}$$

Taking the expectation value of these operators in an eigenstate $|\Phi\rangle$ of the transfer matrix $t(u)$ with corresponding eigenvalue $\Lambda(u)$ we can define an operator $D_N$

$$D_N(\lambda_{n_1}, \ldots, \lambda_{n_2+1})^{\{\alpha\}\{\beta\}} = \frac{\langle\Phi|\tilde{T}_N(\lambda_{n_1}, \ldots, \lambda_{n_2+1})^{\{\alpha\}\{\beta\}}|\Phi\rangle}{\langle\Phi|\Phi\rangle \prod_{k=n_1}^{n_2+1} \Lambda(\lambda_k)}, \tag{23}$$

which by construction is a matrix on the auxiliary space $\simeq \mathcal{V}^N$. Graphically this operator can be depicted as (here shown for $N = 2$ with $n_1 = 1$, $n_2 = 1$):

$$
D_2(\lambda_1, \lambda_2)^{\{\underline{\alpha}\}\{\underline{\beta}\}} =
\begin{array}{c}
\alpha_0 \\ \alpha_1 \\ \alpha_2
\end{array}
\left[ \tilde{T}_2(\lambda_1, \lambda_2) \right]
\begin{array}{c}
\beta_0 \\ \beta_1 \\ \beta_2
\end{array}
\cdot \frac{1}{\langle \Phi | \Phi \rangle \, \Lambda(\lambda_1) \Lambda(\lambda_2)}
$$

$$
=
\begin{array}{c}
\alpha_0 \\ \alpha_1 \\ \alpha_2
\end{array}
\left[
\begin{array}{ccc}
\lambda_1 - u_1 & \cdots & \lambda_1 - u_L \\
\lambda_2 - u_1 & \cdots & \lambda_2 - u_L
\end{array}
\right]
\begin{array}{c}
\beta_0 \\ \beta_1 \\ \beta_2
\end{array}
\cdot \frac{1}{\langle \Phi | \Phi \rangle \, \Lambda(\lambda_1) \Lambda(\lambda_2)} ,
$$

(24)

where the projection onto the eigenstate $|\Phi\rangle$ is indicated by sandwiching of $\tilde{T}_N$. Note that $D_N^{\{\underline{\alpha}\}\{\underline{\beta}\}} = 0$ for $\alpha_{n_1-1} \neq \beta_{n_1-1}$, $\alpha_{n_2+1} \neq \beta_{n_2+1}$ for states $|\Phi\rangle \in \mathcal{H}_{\text{per}}^L$. This allows to decompose $D_N$ into blocks labeled by $\alpha_{n_1-1}$ and $\alpha_{n_2+1}$, i.e.

$$
D_2(\lambda_1, \lambda_2)^{\{\underline{\alpha}\}\{\underline{\beta}\}} = \left( D_2^{[\alpha_0, \alpha_2]}(\lambda_1, \lambda_2) \right)^{\alpha_1}_{\beta_1} ,
$$

(25)

for the example $N = 2$ displayed above.

Comparing to (21) we observe that the reduced density matrix of the face model for $N$ consecutive edges (or segments of the fusion path) in an eigenstate $|\Phi\rangle$ of the transfer matrix is obtained from $D_N$ by proper choice of the arguments $\lambda_k$:

$$
D_N(\lambda_{n_1}, \ldots, \lambda_{n_2+1})^{\{\underline{\alpha}\}\{\underline{\beta}\}}\Big|_{\lambda_k = u_k, \, k = n_1, \ldots, n_2+1} = \frac{1}{\langle \Phi | \Phi \rangle} \langle \Phi | E^{\alpha_{n_1} \ldots \alpha_{n_2}}_{\beta_{n_1} \ldots \beta_{n_2}} | \Phi \rangle .
$$

(26)

In a slight misuse of notation we shall denote $D_N$ as $N$-site density matrix below. $D_N$ is normalized such that $\text{tr}_{\mathcal{V}^N} D_N(\lambda_{n_1}, \ldots \lambda_{n_2+1}) = 1$, which gives a constraint on the diagonal elements of $D_N$. Taking partial traces, i.e. summing over pairs $(\alpha_\ell, \beta_\ell)$ of auxiliary indices, any $n$-point function with $n \leq N$ can be computed from $D_N$.

## 4  Functional equations

For the next step towards the calculation of correlation functions in an IRF model the functional dependence of the density matrices $D_N$ of the generalized problem on the spectral parameters $\lambda_{n_1}, \ldots, \lambda_{n_2+1}$ has to be found. In integrable models such correlation functions are often closely related to solutions of functional equations of quantum Knizhnik-Zamolodchikov type [36]. In the context of face models from the Andrews-Baxter-Forrester (ABF) series [17] such difference equations have been obtained by Foda *et al.* for the infinite lattice using corner transfer matrix and vertex operator techniques [18]. Below we show that the density matrices $D_N$ satisfy a *discrete* version of such equations using the local properties of the Boltzmann weights of an integrable model. Our approach resembles that of Ref. [15] for the Heisenberg model. Other than the approach of Ref. [18], it is also applicable for finite size face models.

To derive the functional equations for the density operators (23) we introduce the linear operator $A_N(\lambda_1, \ldots, \lambda_N) : \text{End}(\mathcal{V}^N) \to \text{End}(\mathcal{V}^N)$. Let $B$ be an arbitrary operator acting on $\mathcal{V}^N$

as defined in (7). The action of $A_N$ on $B$ graphically as

$$\left(A_N(\lambda_1,\ldots,\lambda_N)[B]\right)^{\{\underline{\alpha}\}\{\underline{\beta}\}}$$

$$= \frac{\delta_{\alpha_0\beta_0}\delta_{\alpha_N\beta_N}}{\prod_{i=1}^{N}\rho(\lambda_i-\lambda_N)\rho(\lambda_N-\lambda_i)}\times$$

$$\times \sum_{\{\underline{\gamma}\}\{\underline{\delta}\}} \delta_{\alpha_{N-1}\gamma_N}\prod_{i=1}^{N-1} W\begin{pmatrix}\gamma_{i-1} & \gamma_i \\ \alpha_{i-1} & \alpha_i\end{pmatrix}\lambda_N-\lambda_i\Bigg) B^{\{\underline{\gamma}\}\{\underline{\delta}\}}\times$$

$$\times \prod_{i=1}^{N-1} W\begin{pmatrix}\delta_{i-1} & \beta_{i-1} \\ \delta_i & \beta_i\end{pmatrix}\lambda_i-\lambda_N\Bigg) P_-\begin{pmatrix}\delta_{N-1} & \beta_{N-1} \\ \delta_N & \beta_N\end{pmatrix}$$

$$= \frac{\delta_{\alpha_0\beta_0}\delta_{\alpha_N\beta_N}}{\prod_{i=1}^{N}\rho(\lambda_i-\lambda_N)\rho(\lambda_N-\lambda_i)}\times$$

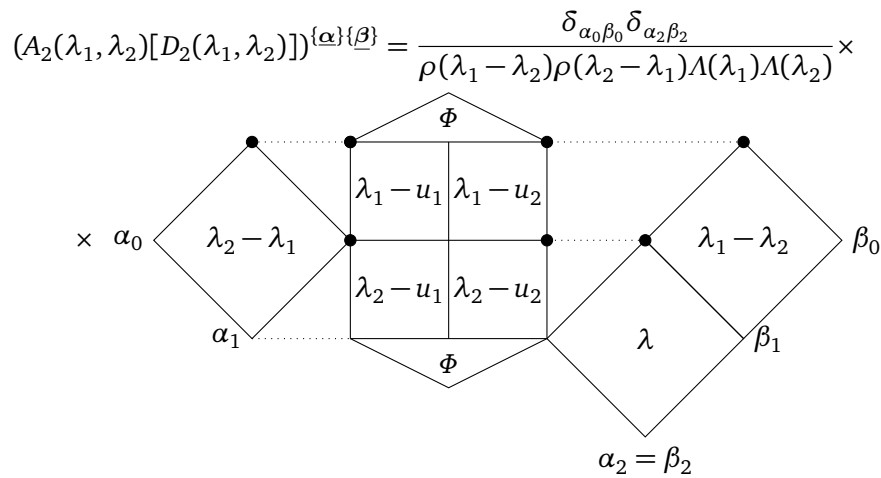

(27)

For models with crossing symmetry as in (15) the operator $P_- \in \mathrm{End}(\mathcal{V}^1)$ is obtained by evaluation of the Boltzmann weight (2) at $u = \lambda$. For more complicated cases this expression needs to be modified, see (46) below. Note the extra Kronecker $\delta$'s enforcing that the image of $B$ has elements acting on $\mathcal{H}_{\mathrm{per}}^L$ only.

As an example consider the action of $A_2$ on the density matrix $D_2$, here shown for a system of length $L = 2$:

$$(A_2(\lambda_1,\lambda_2)[D_2(\lambda_1,\lambda_2)])^{\{\underline{\alpha}\}\{\underline{\beta}\}} = \frac{\delta_{\alpha_0\beta_0}\delta_{\alpha_2\beta_2}}{\rho(\lambda_1-\lambda_2)\rho(\lambda_2-\lambda_1)\Lambda(\lambda_1)\Lambda(\lambda_2)}\times$$

We can now formulate the main theorem of this chapter.

**Theorem 2.** *The density operator $D_N(\lambda_1, \ldots, \lambda_N)$ is a solution of the functional equation*

$$A_N(\lambda_1, \ldots, \lambda_N)[D_N(\lambda_1, \ldots, \lambda_N)] = D_N(\lambda_1, \ldots, \lambda_N + \lambda) \tag{28}$$

*if $\lambda_N$ is equal to one of the inhomogeneities, i.e. $\lambda_N \in \{u_i\}_{i=1}^L$.*

*Proof.* The proof is given in Appendix B. □

For general $\lambda_N$ the functional equation (28) is a difference equation for the elements of the density operator resembling the reduced quantum Knizhnik-Zamolodchikov (qKZ) equation for correlation functions of the six-vertex model in the thermodynamic limit [7, 36, 37]. In general Eq. (28) is valid only for a discrete set of values, namely $\lambda_N \in \{u_1, \ldots, u_N\}$ – as in the finite temperature case for the spin-1/2 Heisenberg model [15]. For the face models considered here, however, it is straightforward to show that this restriction can be dropped for matrix elements of (28) where $\alpha_{N-1}$ is a leaf node on the adjacency graph $\mathfrak{G}$, i.e. if it has exactly one neighbour.

Another functional equation satisfied by the density matrices follows directly from the Yang-Baxter equation. Introducing the operator

$$\langle \underline{\alpha} | W_i(u) | \underline{\beta} \rangle \equiv W \begin{pmatrix} \alpha_{i-1} & \beta_i \\ \alpha_i & \alpha_{i+1} \end{pmatrix} u \Big) \prod_{j \neq i} \delta_{\alpha_j \beta_j}, \tag{29}$$

we have for $1 \leq i < N$:

$$W_i(\lambda_{i+1} - \lambda_i) \cdot D_N(\lambda_1, \ldots, \lambda_i, \lambda_{i+1}, \ldots, \lambda_N) = \\ D_N(\lambda_1, \ldots, \lambda_{i+i}, \lambda_i, \ldots, \lambda_N) \cdot W_i(\lambda_{i+1} - \lambda_i). \tag{30}$$

## 5 Applications

In this section we will use the functional equation (28) to compute the density matrices of face models of solid-on-solid (SOS) type and the related anyon chains. This class of face models has been introduced by Andrews, Baxter and Forrester as an auxiliary tool to solve the 8-vertex model. Specifically, we shall consider two critical models with a finite set $\mathfrak{G}$ of height variables, i.e. the cyclic solid-on-solid (CSOS) model [38, 39] and the restricted solid-on solid (RSOS) model [17].

### 5.1 The cyclic solid-on-solid model

The height variables of the CSOS model take integer values $0 \leq a \leq r-1$ for a positive integer $r$. Heights on adjacent sites are required to differ by $\pm 1$ modulo $r$, hence a configuration of neighbouring spins 0 and $r-1$ is allowed. As a consequence the adjacency graph for this model corresponds to the Dynkin diagram of the affine Lie algebra $\tilde{A}_{r-1}$, see Figure 1(a). The Boltzmann weights of the critical CSOS model are (heights in the arguments of $W$ are taken modulo $r$)

$$\alpha_a = W \begin{pmatrix} a-1 & a \\ a & a+1 \end{pmatrix} u \Big) = W \begin{pmatrix} a+1 & a \\ a & a-1 \end{pmatrix} u \Big) = \frac{\sin(\lambda - u)}{\sin \lambda},$$

$$\beta_a^{\pm} = W \begin{pmatrix} a & a \pm 1 \\ a \mp 1 & a \end{pmatrix} u \Big) = \frac{\sin u}{\sin \lambda},$$

$$\gamma_a = W \begin{pmatrix} a & a+1 \\ a+1 & a \end{pmatrix} u \Big) = 1,$$

$$\delta_a = W \begin{pmatrix} a & a-1 \\ a-1 & a \end{pmatrix} u \Big) = 1. \tag{31}$$

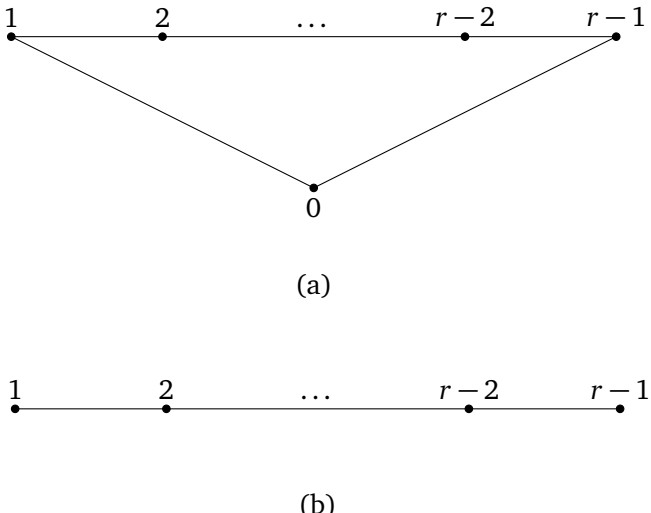

(a)

(b)

Figure 1: Adjacency graphs of the CSOS model (a) and the RSOS model (b).

Here the crossing parameter is $\lambda = \pi m/r$ where $1 \leq m \leq r-1$ is coprime to $r$ (the Boltzmann weights of the general CSOS model are elliptic functions of the spectral parameter $u$ depending explicitly on the height variable $a$ through an additional phase angle which, however, plays no role in the critical case). Note that the weights (31) coincide with the non-zero vertex weights in the $R$-matrix of the six-vertex model. In fact, this relation has been used extensively, e.g. to identify the operator content of the low energy effective theory of the lattice model in the thermodynamic limit [40]. Furthermore, and unlike most other face models, the transfer matrix of the CSOS model has a simple eigenstate which allows for its solution by means of the algebraic Bethe ansatz method based on an $R$-matrix depending on a dynamical parameter related to the height variables. This property has already been used to compute form factors in the basis of Bethe eigenstates of this model [25].

Here we will utilize the existence of a particularly simple eigenstate of the CSOS transfer matrix to illustrate the approach to compute correlation functions based on the functional equation (28). To be specific we choose the CSOS model with $r = 3$ and crossing parameter $\lambda = 2\pi/3$. Considering a lattice of length $L = 3k$ with integer $k$ it is easy to verify that

$$|\Omega\rangle \equiv \frac{1}{\sqrt{3}} \left( |012\,012\ldots0\rangle + |120\,120\ldots1\rangle + |201\,201\ldots2\rangle \right) \in \mathcal{H}^L_{\text{per}} \tag{32}$$

is an eigenstate of the transfer matrix with eigenvalue

$$\Lambda_0(u) = a(u) + d(u), \tag{33}$$

where

$$a(u) \equiv \prod_{i=1}^{L} \frac{\sin(\lambda - (u - u_i))}{\sin \lambda}, \qquad d(u) \equiv \prod_{i=1}^{L} \frac{\sin(u - u_i)}{\sin \lambda}. \tag{34}$$

From the definition of the Boltzmann weights it follows that the action of the single row operators (8) on this state is

$$
\begin{aligned}
T^{\alpha\alpha}_{\alpha+1\alpha+1}(u)|\Omega\rangle &= \frac{a(u)}{\sqrt{3}} |\alpha\,\alpha+1\,\alpha+2\,\alpha\ldots\alpha\rangle, \\
T^{\alpha\alpha}_{\alpha-1\alpha-1}(u)|\Omega\rangle &= \frac{d(u)}{\sqrt{3}} |\alpha\,\alpha-1\,\alpha-2\,\alpha\ldots\alpha\rangle.
\end{aligned}
\tag{35}
$$

This allows to analyze the density matrices $D_N$ in the state (32). The simplest case is $N = 1$ where periodic boundary conditions imply that $\alpha_0 = \beta_0$ and $\alpha_1 = \beta_1$. As a consequence $D_1(\lambda)$ is a diagonal operator on $\mathcal{V}^1$ whose diagonal elements can be directly read off from Eqs. (35), e.g.

$$\langle 01|D_1^{(\Omega)}(\lambda)|01\rangle = \frac{1}{\sqrt{3}}\tilde{a}(\lambda), \quad \langle 02|D_1^{(\Omega)}(\lambda)|02\rangle = \frac{1}{\sqrt{3}}\tilde{d}(\lambda), \tag{36}$$

where $\tilde{a}(u) = a(u)/(\sqrt{3}\,\Lambda_0(u))$ and respectively for $\tilde{d}$. Note that the trace condition $\mathrm{tr}_{\mathcal{V}^1}D_1(\lambda) = 1$ implies $\tilde{d}(\lambda) = 1/\sqrt{3} - \tilde{a}(\lambda)$.

Similarly, the diagonal elements of the two-site density matrix $D_2(\lambda_1, \lambda_2)$ in the reference state are obtained from (35). The functional equation (28) allows for the direct computation of all off-diagonal elements: choosing

$$\{|010\rangle, |012\rangle, |020\rangle, |021\rangle\} \cup \{|121\rangle, |120\rangle, |101\rangle, |102\rangle\} \cup \{|202\rangle, |201\rangle, |212\rangle, |210\rangle\} \tag{37}$$

as a basis for the auxiliary space $\mathcal{V}^2$ and using the fact that the Boltzmann weights are invariant under the shift of all heights by an integer we find that the density matrix has a structure of three identical $4 \times 4$ blocks $D_2^{(\Omega)}(\lambda_1, \lambda_2)$. Restricting ourselves to the first of these blocks we find for the reference state (32)

$$D_2^{(\Omega)}(\lambda_1, \lambda_2) = \begin{pmatrix} \tilde{a}(\lambda_1)\tilde{d}(\lambda_2) & 0 & g(\lambda_1, \lambda_2) & 0 \\ 0 & \tilde{a}(\lambda_1)\tilde{a}(\lambda_2) & 0 & 0 \\ 0 & 0 & \tilde{a}(\lambda_2)\tilde{d}(\lambda_1) & 0 \\ 0 & 0 & 0 & \tilde{d}(\lambda_1)\tilde{d}(\lambda_2) \end{pmatrix}, \tag{38}$$

or, using the notation introduced in Eq. (25),

$$D_2^{(\Omega)[00]}(\lambda_1, \lambda_2) = \begin{pmatrix} \tilde{a}(\lambda_1)\tilde{d}(\lambda_2) & g(\lambda_1, \lambda_2) \\ 0 & \tilde{a}(\lambda_2)\tilde{d}(\lambda_1) \end{pmatrix},$$

$$D_2^{(\Omega)[01]}(\lambda_1, \lambda_2) = \tilde{d}(\lambda_1)\tilde{d}(\lambda_2), \quad D_2^{(\Omega)[02]}(\lambda_1, \lambda_2) = \tilde{a}(\lambda_1)\tilde{a}(\lambda_2). \tag{39}$$

With this ansatz we obtain from the functional equations (28) and (30) after some algebra an explicit expression for the off-diagonal element ($\lambda_{k\ell} \equiv \lambda_k - \lambda_\ell$)

$$g(\lambda_1, \lambda_2) = -\frac{\sqrt{3}}{2\sin(\lambda_{12})}\left(\tilde{d}(\lambda_1)\tilde{a}(\lambda_2) - \tilde{a}(\lambda_1)\tilde{d}(\lambda_2)\right). \tag{40}$$

Hence, as a consequence of the simple form of the reference state (32) of the $r = 3$ CSOS model, the two-site density matrix is completely determined by the one-point function $\tilde{a}(\lambda)$. This is also true for the three-site density matrix $D_3^{(\Omega)}(\{\lambda_1, \lambda_2, \lambda_3\})$. Choosing the bases

$$[0, 0]: |0120\rangle, |0210\rangle, \quad [0, 1]: |0101\rangle, |0121\rangle, |0201\rangle, \quad [0, 2]: |0102\rangle, |0202\rangle, |0212\rangle$$

for the $[0, a]$-blocks in the auxiliary space $\mathcal{V}^3$ we find:[2]

$$D_3^{(\Omega)[0,0]}(\{\lambda_j\}) = \sqrt{3}\begin{pmatrix} \tilde{a}(\lambda_1)\tilde{a}(\lambda_2)\tilde{a}(\lambda_3) & 0 \\ 0 & \tilde{d}(\lambda_1)\tilde{d}(\lambda_2)\tilde{d}(\lambda_3) \end{pmatrix}$$

$$D_3^{(\Omega)[0,1]}(\{\lambda_j\}) = \sqrt{3}\begin{pmatrix} \tilde{a}(\lambda_1)\tilde{d}(\lambda_2)\tilde{a}(\lambda_3) & 0 & -\tilde{a}(\lambda_3)g(\lambda_1, \lambda_2) \\ -\tilde{a}(\lambda_1)g(\lambda_2, \lambda_3) & \tilde{a}(\lambda_1)\tilde{a}(\lambda_2)\tilde{d}(\lambda_3) & \star \\ 0 & 0 & \tilde{a}(\lambda_1)\tilde{a}(\lambda_2)\tilde{a}(\lambda_3) \end{pmatrix} \tag{41}$$

$$D_3^{(\Omega)[0,2]}(\{\lambda_j\}) = \sqrt{3}\begin{pmatrix} \tilde{a}(\lambda_1)\tilde{d}(\lambda_2)\tilde{d}(\lambda_3) & -\tilde{d}(\lambda_1)g(\lambda_1, \lambda_2) & \star\star \\ 0 & \tilde{d}(\lambda_1)\tilde{a}(\lambda_2)\tilde{d}(\lambda_3) & -\tilde{d}(\lambda_1)g(\lambda_2, \lambda_3) \\ 0 & 0 & \tilde{d}(\lambda_1)\tilde{d}(\lambda_2)\tilde{a}(\lambda_3) \end{pmatrix},$$

---

[2]The coefficients in (41) are obtained using a combination of the functional equations (28) and (30) and the algorithm for the calculation of the structure functions in the factorized form of the $N$-site density matrices desscribed below.

with

$$\star = -\frac{\cos(\frac{\pi}{6} - \lambda_{12})\tilde{a}(\lambda_2)g(\lambda_1, \lambda_3)}{\sin\lambda_{12}} - \frac{\sqrt{3}\tilde{a}(\lambda_1)g(\lambda_2, \lambda_3)}{2\sin\lambda_{12}},$$

$$\star\star = \frac{\cos(\frac{\pi}{6} - \lambda_{12})\tilde{d}(\lambda_2)g(\lambda_1, \lambda_3)}{\sin\lambda_{12}} - \frac{\sqrt{3}\tilde{d}(\lambda_1)g(\lambda_2, \lambda_3)}{2\sin\lambda_{12}}.$$

This suffices for the calculation of the nearest and next-nearest neighbour correlation functions in the reference state (32). In the homogeneous limit (i.e. all inhomogeneities $u_k = 0$) we have $\tilde{a}(0) = 1/\sqrt{3}$, $\tilde{d}(0) = 0$, and $g(0,0) = 0$. Therefore, the two and three-site density matrices for $\lambda_i = 0$ are diagonal with non-zero elements

$$\langle 012|D_2^{(\Omega)}(0,0)|012\rangle = \langle 120|D_2^{(\Omega)}(0,0)|120\rangle = \langle 201|D_2^{(\Omega)}(0,0)|201\rangle = \frac{1}{3},$$

$$\langle 0120|D_3^{(\Omega)}(0,0,0)|0120\rangle = \langle 1201|D_3^{(\Omega)}(0,0,0)|1201\rangle = \langle 2012|D_3^{(\Omega)}(0,0,0)|2012\rangle = \frac{1}{3}.$$

(42)

With Eq. (26) this yields the expected results for the two- and three-point functions in the reference state $|\Omega\rangle$ of the $r = 3$ CSOS model.

## 5.2 The restricted solid-on-solid model

The RSOS model can be treated in a similar way. The state space is obtained by removing 0 from the set of heights allowed in the CSOS model. As a consequence the adjacency graph corresponds to the Dynkin diagram of $A_{r-1}$, Fig. 1(b). The Boltzmann weights of the critical RSOS model are again given in terms of trigonometric functions

$$W\begin{pmatrix} a & b \\ c & d \end{pmatrix}u\end{pmatrix} = \delta_{ad}\sqrt{\frac{g_b g_c}{g_a g_d}}\rho(u+\lambda) - \delta_{bc}\rho(u), \tag{43}$$

with

$$\rho(u) = \frac{\sin(u-\lambda)}{\sin\lambda}, \quad g_x = \frac{\sin(\lambda x)}{\sin\lambda} \tag{44}$$

and a crossing parameter $\lambda = \pi/r$. They satisfy the face Yang-Baxter equation (12) and the unitarity condition (14). As a consequence of the gauge factors $g_x$ in the definition of the Boltzmann weights the crossing relation (15) is modified to

$$W\begin{pmatrix} a & b \\ c & d \end{pmatrix}u\end{pmatrix} = \sqrt{\frac{g_b g_c}{g_a g_d}}W\begin{pmatrix} b & d \\ a & c \end{pmatrix}\lambda-u\end{pmatrix}. \tag{45}$$

This modification has to be taken into account whenever crossing symmetry is used, in particular in the definition of the $A$-operator in (27). To cancel the additional factors from the Boltzmann weight evaluated at $u = \lambda$ we have to rescale the corresponding weight giving the operator $P_- \in \text{End}(\mathcal{V}^1)$:

$$\langle\alpha_0\alpha_1\alpha_2|P_-|\beta_0\beta_1\beta_2\rangle = \alpha_1\underset{\alpha_2 = \beta_2}{\overset{\alpha_0 = \beta_0}{\left\langle P_- \right\rangle}}\beta_1 \equiv \delta_{\alpha_0\beta_0}\delta_{\alpha_2\beta_2}\sqrt{\frac{g_{\alpha_0}g_{\alpha_2}}{g_{\alpha_1}g_{\beta_1}}}\,\alpha_1\underset{\alpha_2}{\overset{\alpha_0}{\left\langle \lambda \right\rangle}}\beta_1. \tag{46}$$

In addition, the third step of the proof in Appendix B needs to be reconsidered. Keeping track of the gauge factors we find, that the $A$-operator needs to be multiplied by an additional factor of $\sqrt{g_{\beta_{N-1}}/g_{\alpha_{N-1}}}$.

By construction the transfer matrix (9) of this model and its eigenvalues $\Lambda(u)$ are Fourier polynomials of degree $L$

$$\Lambda(u) = \sum_{n=-L/2}^{L/2} \Lambda_{2n} e^{i2nu} . \tag{47}$$

The leading Fourier coefficients are known to take values [41]

$$\Lambda_{\pm L} = \left( \prod_{\ell=1}^{L} \exp(\mp i(u_\ell + \lambda/2)) \right) \frac{2\cos((2j+1)\lambda)}{(2\sin\lambda)^L} , \quad j \in \{0, \frac{1}{2}, 1, \ldots, \frac{r-2}{2}\} . \tag{48}$$

This allows to decompose the spectrum of the RSOS model into topological sectors with 'quantum dimension'

$$d_q(j) = \frac{\sin(\pi(2j+1)/r)}{\sin(\pi/r)} , \tag{49}$$

labeled by the quantum number $j$. In addition there is a discrete symmetry due to the invariance of the Boltzmann weights under a reflection of all heights, i.e. $a \to r-a$.[3] This symmetry is inherited to the transfer matrix and the reduced density operators.

Starting with the single-site density matrix $D_1(\lambda_1)$ we observe that only its diagonal elements are allowed to be non-zero. We will now prove that $D_1(\lambda)$ is independent of the spectral parameter $\lambda$ in any eigenstate $|\Phi\rangle$ of the transfer matrix (although the matrix elements may still depend on the choice of inhomogeneities $\{u_i\}_{i=1}^{L}$): to compute $D_1^{[12]}(\lambda)$ we note that due to the adjacency condition

$$D_1^{[12]}(\lambda) = \langle 12|D_1(\lambda)|12\rangle = \sum_{\alpha_1} \langle 1\alpha_1|D_1(\lambda)|1\alpha_1\rangle = \frac{\langle\Phi|P_1^{(1)}t(\lambda)|\Phi\rangle}{\langle\Phi|\Phi\rangle\,\Lambda(\lambda)} = \frac{\langle\Phi|P_1^{(1)}|\Phi\rangle}{\langle\Phi|\Phi\rangle} , \tag{50}$$

where we have used the definition of $D_1$. Note that the one-site projection operators, defined as

$$\langle a|P_\ell^{(\bar{a})}|b\rangle = \delta_{a_\ell \bar{a}} \prod \delta_{a_j b_j} , \quad |a\rangle, |b\rangle \in \mathcal{H}^L , \tag{51}$$

are independent of the spectral parameter. Hence, the 1-point function (50) is the local height probability for finding a spin $a = 1$ if $|\Phi\rangle = (\langle\Phi|)^\dagger$. With the same reasoning, one concludes that $D_1^{[21]}(\lambda)$ and the reflected matrix elements are equal to (50). Following the same route, we calculate

$$D_1^{[21]}(\lambda) + D_1^{[23]}(\lambda) = \sum_{\alpha_1} D_1^{[2\alpha_1]}(\lambda) = \frac{\langle\Phi|P_1^{(2)}|\Phi\rangle}{\langle\Phi|\Phi\rangle} . \tag{52}$$

Given that $D_1^{[21]}(\lambda)$ is constant we find that $D_1^{[23]}(\lambda)$ is also independent of $\lambda$. Repeating this procedure we find that in fact all matrix elements are independent of the spectral parameter and given as sums of the local height probabilities. Generically the latter depend on state $|\Phi\rangle$ and the inhomogeneities. For the critical RSOS models considered here we find, however, that they are functions only of $r$ and the local spin in the topological sectors with quantum dimension $d_q = 1$. Using the known values for the local height probabilities in the thermodynamic ground state of the homogeneous system [17] we find

$$D_1^{[a,a+1]}(\lambda) = \frac{\sin(\pi a/r)\sin(\pi(a+1)/r)}{r\cos(\pi/r)} \tag{53}$$

for the non-zero elements of the single-site density matrix in these sectors.

---

[3]Note that this reflection is an automorphism of the underlying fusion algebra $su(2)_{r-2}$ for odd $r$, see the discussion for $r = 5$ in Appendix D. This allows to restrict the possible topological quantum numbers in (48) to take integer values $j \in \{0, 1, \ldots, (r-3)/2\}$ .

**The $r = 4$ RSOS model.** For the simplest nontrivial case, $r = 4$, the height variables take values $1 \leq a \leq 3$, and from the considerations above we immediately get $D_1^{[\alpha\beta]}(\lambda) = \langle \Phi | P_1^{(1)} | \Phi \rangle / \langle \Phi | \Phi \rangle$ for all states $|\alpha\beta\rangle \in \mathcal{V}^1$. Using the trace condition with $\dim \mathcal{V}^1 = 4$ it follows that

$$D_1(\lambda) = \frac{1}{4}\mathbb{1}, \tag{54}$$

independent of the choice of inhomogeneities and agreement with Eq. (53).

For the two-site density matrix we consider the auxiliary space $\mathcal{V}^2$ with dimension six and we choose

$$\{|121\rangle, |123\rangle, |212\rangle, |232\rangle, |321\rangle, |323\rangle)\} \tag{55}$$

as a basis. Similar to the previous reasonings we find

$$\langle 212 | D_2(\lambda_1, \lambda_2) | 212 \rangle = \sum_{\alpha_2} \langle 21\alpha_2 | D_2(\lambda_1, \lambda_2) | 21\alpha_2 \rangle = D_1^{[21]}(\lambda_1) = \frac{1}{4} \tag{56}$$

and also, due to reflection symmetry, $\langle 232 | D_2(\lambda_1, \lambda_2) | 232 \rangle = 1/4$. In addition, we have that

$$\sum_{\alpha_2} \langle 12\alpha_2 | D_2(\lambda_1, \lambda_2) | 12\alpha_2 \rangle = D_1^{[12]}(\lambda_1) = \frac{1}{4}. \tag{57}$$

Hence, we find that the non-zero blocks in $D_2(\lambda_1, \lambda_2)$ are

$$D_2^{[11]}(\lambda_1, \lambda_2) = \frac{1}{8} + \frac{1}{2}f(\lambda_1, \lambda_2), \quad D_2^{[13]}(\lambda_1, \lambda_2) = \frac{1}{8} - \frac{1}{2}f(\lambda_1, \lambda_2),$$
$$D_2^{[22]}(\lambda_1, \lambda_2) = \begin{pmatrix} \frac{1}{4} & g(\lambda_1, \lambda_2) \\ g(\lambda_1, \lambda_2) & \frac{1}{4} \end{pmatrix}, \tag{58}$$

$D_2^{[31]}$ and $D_2^{[33]}$ follow from height reflection $a_i \to r - a_i$. Generically the two functions $f$ and $g$ are independent. Evaluating equation (30) we find that $f(u, v) = f(v, u)$ and $g(u, v) = g(v, u)$, i.e. the two site density operator for $r = 4$ is symmetric under exchange of the arguments.

Taking (58) as an ansatz in the functional equation (28) we obtain $2L$ linear relations for the unknown functions $f$ and $g$ at $\lambda_2 \in \{u_1, u_2, \ldots, u_L\}$:

$$\cos(2\lambda_{12}) f(\lambda_1, \lambda_2 + \lambda) + \sin(2\lambda_{12}) g(\lambda_1, \lambda_2) = \frac{1}{4},$$
$$\cos(2\lambda_{12}) g(\lambda_1, \lambda_2 + \lambda) + \sin(2\lambda_{12}) f(\lambda_1, \lambda_2) = \frac{1}{4}. \tag{59}$$

For the actual computation of of the density matrices we note that, as a consequence of (23) the elements of $D_N(\lambda_1, \ldots, \lambda_N)\prod_{k=1}^{N} \Lambda(\lambda_k)$ are Fourier polynomials in the spectral parameters $\lambda_k$. We have checked that, for small $N$ and system sizes, the $(L+1)^N$ unknown Fourier coefficients can be determined uniquely for a given transfer matrix eigenvalue $\Lambda(u)$ using the discrete functional equations (59) for $N = 2$ and similarly (28) for general $N$ once $D_{N-1}$ is known (cf. the appearance of $D_1$ in the sum rules (56) and (57) for $D_2$).

This procedure is simplified when we consider density operators for eigenstates in the sectors with quantum dimension $d_q(j) = 1$, i.e. topological quantum numbers $j \in \{0, 1\}$: here we find that $D_2$ is determined by a single function of the spectral parameters $f(\lambda_1, \lambda_2) \equiv g(\lambda_1, \lambda_2)$ such that equations (59) for the elements of the two-site density matrix degenerate to a set of $L$ equations. Another simplification in these sectors is found for spectral parameter $\lambda_2 \to i\infty$: in this limit the functions $f$ and $g$ vanish and $D_2(\lambda_1, \lambda_2)$ becomes the single-site density matrix $D_1(\lambda_1)$, written as an operator on $\mathcal{V}^2$ using the basis (55). In fact, we find that a similar reduction relating $D_N$ for large $\lambda_N$ to $D_{N-1}$ for $N \geq 2$ holds in the topological sectors with

quantum dimension $d_q = 1$ of the RSOS models where (recall that $D_1$ is independent of the spectral parameter and diagonal): [4]

$$
\lim_{\lambda_N \to i\infty} [D_N(\lambda_1, \ldots, \lambda_N)]^{\alpha_0 \ldots \alpha_N, \beta_0 \ldots \beta_N}
$$

$$
= [D_{N-1}(\lambda_1, \ldots, \lambda_{N-1})]^{\alpha_0 \ldots \alpha_{N-1}, \beta_0 \ldots \beta_{N-1}} \frac{[D_1]^{\alpha_{N-1}\alpha_N, \beta_{N-1}\beta_N}}{\sum_\alpha [D_1]^{\alpha_{N-1}\alpha, \beta_{N-1}\alpha}} . \tag{60}
$$

Using (30) one obtains expressions for $D_N$ in the limit of large $\lambda_k$, $k < N$.

Hence, the asymptotics of the $N$-site density matrix is determined by the $(N-1)$-site one, e.g. (recall that $f = g$ in these sectors)

$$
\lim_{\lambda_3 \to i\infty} D_3(\lambda_1, \lambda_2, \lambda_3) = \frac{1}{8} \mathbb{1} + \frac{f(\lambda_1, \lambda_2)}{2}
\begin{pmatrix}
1 & 0 & 0 & 0 & 0 & 0 & 0 & 0 \\
0 & -1 & 0 & 0 & 0 & 0 & 0 & 0 \\
0 & 0 & 0 & 1 & 0 & 0 & 0 & 0 \\
0 & 0 & 1 & 0 & 0 & 0 & 0 & 0 \\
0 & 0 & 0 & 0 & 0 & 1 & 0 & 0 \\
0 & 0 & 0 & 0 & 1 & 0 & 0 & 0 \\
0 & 0 & 0 & 0 & 0 & 0 & -1 & 0 \\
0 & 0 & 0 & 0 & 0 & 0 & 0 & 1
\end{pmatrix}, \tag{61}
$$

for the three-site density matrix of the $r = 4$ model in the basis

$$
\{|1212\rangle, |1232\rangle\} \cup \{|2121\rangle, |2321\rangle\} \cup \{|2123\rangle, |2323\rangle\} \cup \{|3212\rangle, |3232\rangle\}
$$

of $\mathcal{V}^3$.

Remarkably, it has been shown that the density matrices of the Heisenberg spin chain can be written as

$$
D_N(\lambda_1, \ldots, \lambda_N) = \sum_{m=0}^{[N/2]} \sum_{I,J} \left( \prod_{p=1}^{m} \omega(\lambda_{i_p}, \lambda_{j_p}) \right) f_{N;I,J}(\lambda_1, \ldots, \lambda_N), \tag{62}
$$

in terms of a nearest neighbour two-point function $\omega$ and a set of recursively defined elementary functions $f_{N;I,J}$ of the spectral parameters $\lambda_j$, so-called 'structure functions' [7]. Here $I = (i_1, \ldots, i_m)$ and $J = (j_1, \ldots, j_m)$ such that $I \cap J = \emptyset$, $1 \le i_p < j_p \le N$ and $i_1 < \cdots < i_m$.

For the density matrices in eigenstates from the topological sectors with quantum dimension $d_q(j) = 1$ ($j \in \{0, 1\}$ for the $r = 4$ RSOS model) we observe a similar behaviour, e.g. for the three-point density matrix: motivated by Eq. (62) we assume that the matrix elements of $D_3(\lambda_1, \lambda_2, \lambda_3)$ can be written as

$$
\begin{aligned}
f_0(\lambda_1, \lambda_2, \lambda_3) &+ f_{1,2}(\lambda_1, \lambda_2, \lambda_3) f(\lambda_1, \lambda_2) \\
&+ f_{2,3}(\lambda_1, \lambda_2, \lambda_3) f(\lambda_2, \lambda_3) + f_{1,3}(\lambda_1, \lambda_2, \lambda_3) f(\lambda_1, \lambda_3),
\end{aligned} \tag{63}
$$

where $f_0$ and the $f_{I,J}$ are rational functions of $e^{2i\lambda_{12}}$ and $e^{2i\lambda_{23}}$ ($\lambda_{k\ell} \equiv \lambda_k - \lambda_\ell$), and $f(u, v)$ is the single function from (58) which determines the two-site density matrix in these topological sectors.

Most importantly, the model parameters such as the system size $L$ and the inhomogeneities $\{u_k\}$ enter the expressions (62) only via the two-point function $\omega$ (or $f$ in (63)). This fact can be used to implement an efficient algorithm[5] for the numerical calculation of $f_0$ and $f_{I,J}$ in the ansatz (63) for the 3-site density matrix of the $r = 4$ RSOS model (or the structure functions $f_{N;I,J}$ appearing in an ansatz such as (62) for elements of the $N$-site density matrix $D_N$):

---

[4]A similar reduction has been found to be satisfied by the density matrices of the Heisenberg spin chain [14].

[5]A similar method has been used to compute expectation values of local operators for the spin-1/2 Heisenberg chain in a particular basis [42–44].

1. choose a set of spectral parameters $\Lambda = \{\lambda_1, \dots, \lambda_N\}$,

2. diagonalize the transfer matrix of a sufficiently small system with randomly chosen inhomogeneities,

3. pick an eigenstate of the transfer matrix (from the topological sector considered) and compute the generalized $N$-site density matrix $D_N(\lambda_1, \dots, \lambda_N)$ and the two-site density matrix $D_2(\lambda_j, \lambda_k)$ for pairs $(\lambda_j, \lambda_k)$ from $\Lambda$ using their definition (23),

4. compare $D_2$ to (58) to obtain numerical values for the corresponding two-point functions $f(\lambda_j, \lambda_k)$,

5. insert the data from steps 3 and 4 into (63) (resp. (62)) to get a linear equation relating the structure functions,

6. repeat steps 2 to 5 to build a system of linear equations which can be solved for the structure functions $f_{N;I,J}(\lambda_1, \dots, \lambda_N)$.

Once these functions are known for a range of spectral parameters it is straightforward to find analytical expressions, e.g. by Fourier analysis, which can be checked using (28).

A slight complication in the present case of the $r = 4$ RSOS model is that the decomposition (63) is not unique. Evaluating the diagonal element $\underline{\alpha} = \underline{\beta} = (1, 2, 1, 2)$ of the functional equation (28) for $D_3(x, y, z)$ we find an additional relation satisfied by the two-point function $f$:

$$\sin(2\lambda_{12}) f(\lambda_1, \lambda_2) + \sin(2\lambda_{23}) f(\lambda_2, \lambda_3) - \sin(2\lambda_{13}) f(\lambda_1, \lambda_3) = 0. \tag{64}$$

This identity holds for arbitrary values of $\lambda_j$, $j = 1, 2, 3$, as a consequence of $\alpha_2 = 1$ being a leaf node on the adjacency graph (c.f. the remark in Appendix B).

Taking this into account we have used the procedure outlined above to compute the factorized form of the three-point density matrix $D_{N=3}$ of the $r = 4$ RSOS model. Remarkably, it turns out to be sufficient to compute the initial data for a system of length $L = 2 < N$. Moreover, we find that the structure functions are the same for all eigenstates $|\Phi\rangle$ of the transfer matrix in the topological sectors considered here. As a result we obtain

$$
\begin{aligned}
D_3^{[12]}(\lambda_1, \lambda_2, \lambda_3) &= \frac{1}{8} \mathbb{1} + \frac{1}{2\sin(2\lambda_{23})} \begin{pmatrix} \sin(2\lambda_{23}) & -1 \\ 1 & -\sin(2\lambda_{23}) \end{pmatrix} f(\lambda_1, \lambda_2) \\
&\quad + \frac{\cos(2\lambda_{23})}{2\sin(2\lambda_{23})} \begin{pmatrix} 0 & 1 \\ -1 & 0 \end{pmatrix} f(\lambda_1, \lambda_3) + \frac{1}{2} \begin{pmatrix} 0 & 1 \\ 1 & 0 \end{pmatrix} f(\lambda_2, \lambda_3), \\
D_3^{[21]}(\lambda_1, \lambda_2, \lambda_3) &= \frac{1}{8} \mathbb{1} + \frac{1}{2} \begin{pmatrix} 0 & 1 \\ 1 & 0 \end{pmatrix} f(\lambda_1, \lambda_2) + \frac{\cos(2\lambda_{12})}{2\sin(2\lambda_{12})} \begin{pmatrix} 0 & 1 \\ -1 & 0 \end{pmatrix} f(\lambda_1, \lambda_3) \\
&\quad + \frac{1}{2\sin(2\lambda_{12})} \begin{pmatrix} \sin(2\lambda_{12}) & -1 \\ 1 & -\sin(2\lambda_{12}) \end{pmatrix} f(\lambda_2, \lambda_3).
\end{aligned} \tag{65}
$$

As before the other non-zero blocks follow from the height reflection symmetry of the density matrix. These expressions are unique up to transformations based on Eq. (64).

Again we can consider the homogeneous limit $u_k \equiv 0$ for $k = 1, \dots, L$ where the expectation values of $N$-point functions can be obtained from the density matrix

$$D_N(\lambda_1, \dots, \lambda_N)|_{\lambda_k \equiv 0}, \tag{66}$$

according to Eq. (26). In this case the one-point function $D_1(0)$ is already given by (54). In addition, $D_2(0, 0)$ is completely fixed by the two-point function $f(0, 0)$.

For the computation of $D_3(0,0,0)$ the singularities for $\lambda_1 = \lambda_2$ and $\lambda_2 = \lambda_3$ in Eq. (65) has to be taken care of. We expand the two-point function as

$$f(\lambda_1, \lambda_2) \simeq (0,0) + (1,0)\lambda_1 + (0,1)\lambda_2 + \frac{1}{2}\left((2,0)\lambda_1^2 + 2(1,1)\lambda_1\lambda_2 + (0,2)\lambda_2^2\right) + \dots, \quad (67)$$

with $(k,\ell) \equiv \partial_1^k \partial_2^\ell f(\lambda_1, \lambda_2)|_{\lambda_1 = \lambda_2 = 0}$. Note that $(k,\ell) = (\ell,k)$ due to the symmetry of $f(\lambda_1, \lambda_2)$. Additional relations between the coefficients of the $r = 4$ two-point function follow from the identity (64), e.g. $(1,0) = 0$ and $(2,0) - 2(1,1) = 4(0,0)$. As a result the singularities are removed and the homogeneous limit of $D_3$ is found to be

$$D_3(\lambda_1, \lambda_2, \lambda_3)|_{\lambda_k \equiv 0} = \frac{1}{8}\mathbb{1} + \frac{f(0,0)}{2}\begin{pmatrix} 1 & 1 & 0 & 0 & 0 & 0 & 0 & 0 \\ 1 & -1 & 0 & 0 & 0 & 0 & 0 & 0 \\ 0 & 0 & -1 & 1 & 0 & 0 & 0 & 0 \\ 0 & 0 & 1 & 1 & 0 & 0 & 0 & 0 \\ 0 & 0 & 0 & 0 & 1 & 1 & 0 & 0 \\ 0 & 0 & 0 & 0 & 1 & -1 & 0 & 0 \\ 0 & 0 & 0 & 0 & 0 & 0 & -1 & 1 \\ 0 & 0 & 0 & 0 & 0 & 0 & 1 & 1 \end{pmatrix}. \quad (68)$$

As a consequence the two- and three-point correlations in a transfer matrix eigenstate $|\Phi\rangle$ from the topological sectors with $d_q = 1$ of the homogeneous $r = 4$ RSOS model are given in terms of $(0,0)$, i.e. the numerical value of the two-point function at spectral parameters $\lambda_1 = \lambda_2 = 0$, alone. The latter is directly related to the corresponding eigenvalue of the RSOS hamiltonian $H = J \partial_u \ln t(u)|_{u=0}$, i.e.

$$E_\Phi = 4J L f(0,0). \quad (69)$$

Hence, explicit expressions for two- and three-point functions in the ground states of the infinite system can be obtained from Eqs. (58) and (68) using the known results for the energy density of the RSOS model in the thermodynamic limit [45]. We find

$$f(0,0) = \pm\frac{1}{2\pi}, \quad (70)$$

for the ground state of the RSOS hamiltonian with $J = -1$ (+1).

**The $r = 5$ RSOS model.** As a second example we consider the $r = 5$ RSOS model with local heights $1 \le a \le 4$. Similarly as above, we can compute the single site density matrix. In states from the sector with topological quantum number $j = 0$ (recall that the topological sectors in the odd $r$ RSOS models are labelled by integers $0 \le j \le (r-3)/2 = 1$) we find that the matrix elements are independent of the system size and the inhomogeneities $\{u_k\}$, namely

$$D_1^{[\alpha\beta]}(\lambda) = \begin{cases} 1/(5 + \sqrt{5}) & \text{for } (\alpha\beta) \in \{(12),(21),(34),(43)\}, \\ \sqrt{5}/10 & \text{for } (\alpha\beta) \in \{(23),(32)\}, \end{cases} \quad (71)$$

as given by Eq. (53).

The auxiliary space $\mathcal{V}^2$ for the two-site density matrix of the $r = 5$ model has dimension ten. Due to the adjacency condition the Hilbert space of states splits into two spanned by fusion paths with $a_0$ even and odd, respectively. The transfer matrix is a map between these two subspaces. Similarly, products of an even number of transfer matrices (or more general single row operators) are therefore block diagonal and may be written as the sum of an even and and odd part. In view of this decomposition of the Hilbert space we chose the following basis for $\mathcal{V}^2$:

$$\{|121\rangle, |123\rangle, |321\rangle, |323\rangle, |343\rangle\} \cup \{|212\rangle, |232\rangle, |234\rangle, |432\rangle, |434\rangle\}. \quad (72)$$

The two sets are related by reflection $a_i \rightarrow r - a_i$ and hence we may restrict ourselves to the subspace generated by the first. Again the structure of the density operator $D_2(\lambda_1, \lambda_2)$ is constrained by sum rules such as (56) and (57). We find the non-zero blocks of $D_2$ in the first subspace with odd $a_0$ to be ($b$ is a constant)

$$
\begin{aligned}
D_2^{[1,1]}(\lambda_1, \lambda_2) &= -\frac{1}{2} + 4D_1^{[21]} + f(\lambda_1, \lambda_2), \\
D_2^{[1,3]}(\lambda_1, \lambda_2) &= D_2^{[3,1]}(\lambda_1, \lambda_2) = e(\lambda_1, \lambda_2), \\
D_2^{[3,3]}(\lambda_1, \lambda_2) &= \begin{pmatrix} d(\lambda_1, \lambda_2) & c_1(\lambda_1, \lambda_2) \\ c_2(\lambda_1, \lambda_2) & b \end{pmatrix}.
\end{aligned}
\tag{73}
$$

The sum rules immediately imply

$$
e(\lambda_1, \lambda_2) = \frac{1}{2} - 3D_1^{[21]} - f(\lambda_1, \lambda_2), \quad b = D_1^{[21]}.
\tag{74}
$$

Furthermore the trace condition $\mathrm{tr}_{\mathcal{V}^2} D_2(\lambda_1, \lambda_2) = 1$ yields

$$
d(\lambda_1, \lambda_2) = f(\lambda_1, \lambda_2) + D_1^{[21]}.
\tag{75}
$$

Using the relations (30) find that the off-diagonal functions are related via $c_1(\lambda_1, \lambda_2) = c_2(\lambda_2, \lambda_1) \equiv (\sqrt{5} + 2)^{\frac{1}{2}} g(\lambda_1, \lambda_2)$ and

$$
f(\lambda_1, \lambda_2) = \frac{g(\lambda_1, \lambda_2) + g(\lambda_2, \lambda_1)}{2} + \left(5 + 2\sqrt{5}\right)^{\frac{1}{2}} \cot \lambda_{12} \frac{g(\lambda_1, \lambda_2) - g(\lambda_2, \lambda_1)}{2}.
\tag{76}
$$

We find further simplifications for eigenstates of the transfer matrix belonging to the $j = 0$ topological sector (where $b = 1/(5 + \sqrt{5})$ according to Eq. (71)): in this sector the off-diagonal elements of $D_2$ coincide, i.e. $g(\lambda_1, \lambda_2) = g(\lambda_2, \lambda_1)$, and therefore $g(\lambda_1, \lambda_2) = f(\lambda_1, \lambda_2)$. As a consequence $D_2$ can again be expressed in terms of a single scalar function $f(\lambda_1, \lambda_2)$ satisfying the functional equation

$$
f(\lambda_1, \lambda_2 + \lambda) = \frac{1}{5 + 3\sqrt{5} \cos(2\lambda_{12})} + \frac{\cos(2(\lambda_{12} - \lambda)) - \cos(2\lambda)}{\cos(2\lambda_{12}) - \cos(2\lambda)} f(\lambda_1, \lambda_2),
\tag{77}
$$

for $\lambda_2 \in \{u_1, \dots, u_L\}$. As in the $r = 4$ RSOS model the density matrices $D_N$ can be computed recursively for any given transfer matrix eigenvalue using their analytical properties and the functional equations. In particular, we find that the asymptotic behaviour of the $N$-site density matrices is related to the $N - 1$-site one as given by (60), e.g.

$$
\lim_{\lambda_2 \rightarrow i\infty} D_2^{(\mathrm{odd,odd})}(\lambda_1, \lambda_2) = \frac{1}{2(\sqrt{5} + 5)} \begin{pmatrix} 3 - \sqrt{5} & 0 & 0 & 0 & 0 \\ 0 & \sqrt{5} - 1 & 0 & 0 & 0 \\ 0 & 0 & \sqrt{5} - 1 & 0 & 0 \\ 0 & 0 & 0 & 2 & 0 \\ 0 & 0 & 0 & 0 & 2 \end{pmatrix},
\tag{78}
$$

in the topological sector with quantum dimension $d_q = 1$, i.e. $j = 0$ for the $r = 5$ RSOS model.

Similar as in (65) for $r = 4$ we have been able to express the three-point density matrix of the $r = 5$ RSOS model in this topological sector as a sum of terms factorizing into spectral-parameter dependent elementary functions and the two-point function $f(\lambda_1, \lambda_2)$ solving the functional equation (77). Proceeding as for $r = 4$ we find the factorization of the $D_3$ in the one-dimensional block corresponding to the sequence $\underline{\alpha} = (1234)$ of heights to be

$$
\begin{aligned}
D_3^{[14]}(\lambda_1, \lambda_2, \lambda_3) = {} & \frac{7}{4\sqrt{5}} - \frac{3}{4} - \frac{1}{4}\left(\sqrt{5} + 1 + (3\sqrt{5} - 5)\cot \lambda_{13} \cot \lambda_{23}\right) f(\lambda_1, \lambda_2) \\
& - \frac{1}{4}\left(\sqrt{5} + 1 + (3\sqrt{5} - 5)\cot \lambda_{12} \cot \lambda_{13}\right) f(\lambda_2, \lambda_3) \\
& - \frac{1}{4}\left(\sqrt{5} + 1 - (3\sqrt{5} - 5)\cot \lambda_{12} \cot \lambda_{23}\right) f(\lambda_1, \lambda_3).
\end{aligned}
\tag{79}
$$

We present the complete list of non-zero matrix elements of $D_3$ in Appendix C.

Now it is straightforward, to calculate $D_3$ in the homogeneous limit. Expanding the two-point function as in Eq. (67) for the case $r = 4$ we find for the one-dimensional block considered above

$$D_3^{[14]}(0,0,0) = \frac{7}{4\sqrt{5}} - \frac{3}{4} - 2(0,0) + \frac{1}{8}\left(3\sqrt{5} - 5\right)\left(2(1,1) - (2,0)\right). \tag{80}$$

All other matrix elements may be computed using Appendix C.

### 5.3 Fibonacci anyons

As discussed earlier we can relate face models to one-dimensional quantum chains with anyonic degrees of freedom on each lattice site. Considering the Hamiltonian limit of the homogeneous RSOS model with $r = 5$, i.e. $u_i \equiv 0$ in (8), one obtains an integrable model of $su(2)_3$ or Fibonacci anyons [28, 46]. Despite its simplicity this non-Abelian anyon model gives rise to universal quantum computation. It contains only two types of anyons, the trivial anyon 1 and a second one, $\tau$. Here we will use the functional equation (28) to compute the two-site density matrix for the chain of $\tau$-anyons.

The Hilbert space of fusion paths for these anyons can be shown to be isomorphic to the $a_0$ odd part of the RSOS Hilbert space $\mathcal{H}_{\text{per}}^L$ for $r = 5$. A Hamiltonian for a chain of $L$ $\tau$-anyons with local interaction can be constructed using the operators $P^{(\tau\tau\to1)} = \mathbb{1} - P^{(\tau\tau\to\tau)}$ projecting on one of the outcomes of fusing neighbouring anyons according to the rule $\tau \otimes \tau = 1 \oplus \tau$. In Appendix D we show that these operators can be expressed in terms of the Boltzmann weights of the RSOS model (43). This allows for an embedding of the anyon Hamiltonian

$$H = J \sum_{n=0}^{L-1} P_n^{(\tau\tau\to1)} \tag{81}$$

into the family of commuting operators generated by the transfer matrix of the RSOS transfer matrix (9). By choosing a negative (positive) coupling constant $J$ fusion of two neighbouring anyons to a trivial ($\tau$) anyon is energetically favoured.

We will now use the inverse problem and relate the energies of the anyon model to the density operator of the homogeneous model. Note that in this case $t(0)$ is the translation operator with eigenvalues $\Lambda(0) = \exp(2\pi i k/L)$ for some integer $k$. Furthermore, we have $\rho(0) = -1$ for the RSOS model. Assuming that the eigenstates of the transfer matrix are normalized, $\langle\Phi|\Phi\rangle = 1$, the two-point function (26) is

$$\langle\Phi|E_{\beta_0\beta_1\beta_2}^{\alpha_0\alpha_1\alpha_2}|\Phi\rangle = D_2(0,0)^{\{\alpha\}\{\beta\}}. \tag{82}$$

Translation invariance, i.e $[H, t(0)] = 0$, implies that the energy is $E_\Phi = LJ\langle\Phi|P_1^{(\tau\tau\to1)}|\Phi\rangle$ for any eigenstate $|\Phi\rangle$. As $P_1^{(\tau\tau\to1)}$ depends only on the first three heights of the chain, we can directly use (82) and obtain

$$E_\Phi = LJ\,\text{tr}_{\mathcal{V}^2}\left(P_1^{(\tau\tau\to1)}D_2(0,0)\right), \tag{83}$$

relating the energy density of the anyon chain to certain correlators of the RSOS model.

Plugging in the the explicit expression of $P_1^{(\tau\tau\to1)}$ into (83) and using the simplified form of the two-site density matrix (73) for eigenstates $|\Phi\rangle$ in the topological sector $j = 0$ of the $r = 5$ RSOS model [6] we finally obtain

$$\frac{E_\Phi}{L} = \frac{J}{2}\left(\sqrt{5} + 5\right)(0,0) + \frac{J}{2}\left(3 - \sqrt{5}\right). \tag{84}$$

---

[6]This condition holds in particular for the ground states of the antiferromagnetic anyon chain ($J > 0$) with $L$ mod $3 = 0$ and the ferromagnetic model ($J < 0$).

As for the $r = 4$ RSOS model the ground state energies of the anyon model in the thermodynamic limit are known [45] giving

$$(0,0) = \begin{cases} -2 + \sqrt{5} + \frac{1}{3}\sqrt{\frac{5}{6}(25 - 11\sqrt{5})} & \text{for } J > 0 \\ 1 - 2/\sqrt{5} & \text{for } J < 0 \end{cases}, \tag{85}$$

for the corresponding two-point functions $f(0,0)$. Finally, we show how our results can be used for the computation of 3-point functions. Therefore, we consider the operator $P^{(\tau\tau\tau\to 1)}$ which projects the fusion product of three consecutive $\tau$-anyons to an anyon of type 1. Using the homogeneous limit of $D_3$ (again for eigenstates $|\Phi\rangle$ in the topological sector $j = 0$ of the $r = 5$ RSOS model) as discussed in the previous section and (26) we find

$$\langle\Phi|P_1^{(\tau\tau\tau\to 1)}|\Phi\rangle = \sqrt{5} - 2 - 2\left(\sqrt{5} + 5\right)(0,0) - \frac{5}{4}\left(\sqrt{5} - 1\right)\left(2(1,1) - (2,0)\right). \tag{86}$$

## 6 Conclusion

We have studied correlation functions for generic models with interactions-round-a-face on a square lattice (and the related anyonic quantum chains). To make use of the Yang-Baxter integrability local operators have been expressed in terms of (generalized) transfer matrices for inhomogeneous versions of these models, see Eqs. (18) and (21). This allowed to encode correlation functions in $N$-point reduced density matrices $D_N$, Eq. (23), depending on a set of $N$ spectral parameters. We have constructed a set of discrete functional equations of reduced quantum Knizhnik-Zamolodchikov type (28) which determine the functional dependence of $D_N$ on these spectral parameters.

This framework has been applied to several critical solid-on-solid models: in the simple 'reference' state (32) of the $r = 3$ CSOS model we have obtained explicit expressions for the generalized reduced density matrices on up to three neighbouring sites in terms of the one-point function depending on the spectral parameter and the choice of inhomogeneities. By contrast, the one-point functions in the RSOS models are independent of the spectral parameter. For the $r = 4$ and 5 RSOS models we have been able to express the two-site density matrices in terms of two unknown functions similar to the spin-1/2 Heisenberg chains [13, 15]. We find that these functions (and, together with the application of the sum rules, the elements of the $N > 2$-site density matrices) are uniquely determined by the discrete functional equations for any transfer matrix eigenstate, given the analytical properties inherited from the Boltzmann weights.

Additional properties of the density matrices are found in topological sectors with quantum dimension $d_q = 1$ (containing the ground states of the RSOS model): here we have observed a reduction relating $D_N$ to $D_{N-1}$ when one of the spectral parameters is sent to infinity. This resembles the asymptotic condition on the density matrices complementing the discrete qKZ equation for the Heisenberg spin chain guaranteeing the uniquess of its solution [14, 15]. Moreover, we have found that the reduced density operators of the $r = 4$ and 5 RSOS models in these topological sectors can be expressed in terms of a single function determining the nearest-neighbour two-site correlations in the particular transfer matrix eigenstate together with a set of elementary structure functions. This observation and preliminary results for $r > 5$ lead us to conjecture that this holds for all RSOS models in these topological sectors.

For the remaining tasks of calculation of the nearest-neighbour function (the *physical* part of the correlation functions) and the structure functions (the *algebraic* part) in the topological sectors with $d_q = 1$ we have used an ansatz motivated by the 'factorized' form of density matrices for the spin-1/2 Heisenberg chain [7] together with the observation that the density

operator depends on the particular realization of the model, i.e. the system size and choice of inhomogeneities, only via the two-site function. This allows for an efficient computation of the structure functions. We have applied this algorithm to reduce the calculation of the density operators on three neighbouring sites for the $r = 4$ and $r = 5$ RSOS models and related correlation functions for the Fibonacci anyon chain to that of the physical part. The latter solves discrete difference equations (59) and (77) resulting from the functional equation for the two-site density matrix. Explicit expressions for the two-site functions solving these equations are limited to RSOS models of sufficiently small length or a special state as for the CSOS model. In the fermionic basis approach for the spin-1/2 Heisenberg model the physical part of the correlation functions at both finite length and finite temperature has been characterized in terms of integral formulae and difference equations [13] or in terms of solutions to linear and non-linear integral equations [47]. For face models such representation of the two-site function is not known. As a first step, one might consider the zero temperature ground state of these models in the thermodynamic limit: choosing a continuous distribution of the inhomogeneities the functional equations (28) are expected to hold for arbitrary complex values of the spectral parameter $\lambda_N$ allowing for their solution.

To conclude let us mention just two open problems which may be addressed based on the approach presented here: first of all, in the context of RSOS models a comparison of the density matrices with the corresponding quantities for the related anisotropic Heisenberg chains at roots of unity can provide insights on the boundary contributions to correlation functions resulting from the peculiar fusion path nature of the RSOS Hilbert space. Secondly we want to emphasize that the discrete functional equations (28) for the density operators hold for generic integrable IRF models (such equations are also known for vertex models and spin chains related to quantum groups [48]). Together with the algorithm used here for the computation of the structure functions in factorized expressions (62) and (63) this may well allow to shed some light on the question whether the factorization of correlation functions is a general property of integrable models which extends beyond RSOS models and spin-1/2 chains.

# Acknowledgements

We would like to thank H. Boos, F. Göhmann, and A. Klümper for stimulating discussions. This work is part of the programme of the research unit *Correlations in Integrable Quantum Many-Body Systems* (FOR 2316). Partial Funding has been provided by the Deutsche Forschungsgemeinschaft under grant no. FR 737/8.

# A    Proof of Theorem 1

Here we provide the proof of (18) for arbitrary values of $L$ and $1 \leq n < L$. Let us first look at its graphical representation (double arrows indicating periodical boundary conditions in the

horizontal direction): for $|a\rangle, |b\rangle \in \mathcal{H}_{\text{per}}^L$ we consider the matrix element

$$\langle a| \left( \prod_{k=1}^{n-1} t(u_k) \right) T_{\alpha\beta}(u_n) T^{\alpha\beta}(u_{n+1}) \left( \prod_{k=n+2}^{L} t(u_k) \right) |b\rangle =$$

$$= \alpha \begin{array}{cccccc} a_0 & a_1 & a_2 & a_{L-2} & a_{L-1} & a_L \\ 0 & u_{1,2} & \cdots & u_{1,L-1} & u_{1,L} \\ u_{2,1} & 0 & \cdots & u_{2,L-1} & u_{2,L} \\ \vdots & \vdots & \vdots & \vdots & \vdots \\ u_{n,1} & u_{n,2} & \cdots & u_{n,L-1} & u_{n,L} \\ u_{n+1,1} & u_{n+1,2} & \cdots & u_{n+1,L-1} & u_{n+1,L} \\ \vdots & \vdots & \vdots & \vdots & \vdots \\ u_{L-1,1} & u_{L-1,2} & \cdots & 0 & u_{L-1,L} \\ u_{L,1} & u_{L,2} & \cdots & u_{L,L-1} & 0 \\ b_0 & b_1 & b_2 & b_{L-2} & b_{L-1} & b_L \end{array} \beta \quad . \qquad (A.1)$$

After using the initial condition (16) in each row the Boltzmann weights can be turned into Kronecker $\delta$'s by repeated use of unitarity (14). To understand the principle we have a closer

look at the four rows $n-1$, $n$, $n+1$ and $n+2$, i.e.

$$\cdots t(u_{n-1})T_{\alpha\beta}(u_n)T^{\alpha\beta}(u_{n+1})t(u_{n+2})\cdots =$$

$$= \rho_{n-1,n}\rho_{n,n+1}\rho_{n+1,n+2}\times$$

$$= \rho_{n-1,n}\rho_{n,n+1}\rho_{n+1,n+2}\rho_{n-1,n+1}\rho_{n,n+2}\times$$

$$= \rho_{n-1,n}\rho_{n,n+1}\rho_{n+1,n+2}\rho_{n-1,n+1}\rho_{n,n+2}\rho_{n-1,n+2}\times$$

where we used $u_{i,j} \equiv u_i - u_j$ and $\rho_{i,j} \equiv \rho(u_i - u_j)\rho(u_j - u_i)$. Continuing this procedure throughout (A.1) as on the shaded regions shown above, it is easy to see that

$$\langle a| \left( \prod_{k=1}^{n-1} t(u_k) \right) T_{\alpha\beta}(u_n) T^{\alpha\beta}(u_{n+1}) \left( \prod_{k=n+2}^{L} t(u_k) \right) |b\rangle =$$

$$= \prod_{k,\ell=1}^{L} \rho(u_k - u_\ell)$$

$$= \prod_{k,\ell=1}^{L} \rho(u_k - u_\ell) \delta_{a_n,\alpha} \delta_{b_n,\beta} \prod_{j\neq n} \delta_{a_j b_j}$$

$$= \prod_{k,\ell=1}^{L} \rho(u_k - u_\ell) \langle a| \left( E_\beta^\alpha \right)_n |b\rangle$$

is produced which finishes the proof.

## B  Proof of Theorem 2

Here we will provide the proof for the functional equation (28). It consists of three steps. The idea is, to consider the action of $A_N$ on a density operator with one row added, i.e. $D_{N+1}(\lambda_1, \ldots, \lambda_N, \lambda_N + \lambda)$. Recall, that for every $n \in \mathbb{N}$ $D_n(\lambda_1, \ldots, \lambda_n)^{\{\alpha\}\{\beta\}} = 0$ if $\alpha_0 \neq \beta_0$ or $\alpha_n \neq \beta_n$. In those cases the functional equation holds trivially, so we may assume $\alpha_0 = \beta_0$. Also note that $(A_N(\lambda_1, \ldots, \lambda_N)[D_{N+1}(\lambda_1, \ldots, \lambda_N, \lambda_N + \lambda)])^{\{\alpha\}\{\beta\}}$ is a matrix element of an operator $\mathcal{V}^{N+1} \to \mathcal{V}^{N+1}$. To keep the presentation legible, the following steps will be shown

graphically for $N = L = 2$, e.g.

$$A_2(\lambda_1, \lambda_2)[D_3(\lambda_1, \lambda_2, \lambda_2 + \lambda)]^{\{\underline{\alpha}\}\{\underline{\beta}\}} = \frac{\delta_{\alpha_0\beta_0}\delta_{\alpha_2\beta_2}}{\rho(\lambda_1 - \lambda_2)\rho(\lambda_2 - \lambda_1)\Lambda(\lambda_1)\Lambda(\lambda_2)\Lambda(\lambda_2 + \lambda)} \times$$

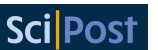

Now, writing $\mathcal{V}^{N+1} = \mathcal{V}^N \hat{\otimes} \mathcal{V}^1$ (see Section 2 for the definition of the symbol $\hat{\otimes}$) we perform two '(constrained) partial traces' over the factor $\mathcal{V}^1$ each leading to one side of the functional equation, i.e. operators on $\mathcal{V}^N$. In a final step we show that for the special choice of $\lambda_N$ being one of the inhomogeneities $\{u_i\}$ the constraint can be dropped and both summations lead to the same result.

In a first step we note that from the definition (23) of the density operators

$$\sum_{\alpha_{N+1}} D_{N+1}(\lambda_1, \dots \lambda_N, \lambda_N + \lambda)^{\{\underline{\alpha}\}\{\underline{\beta}\}} = D_N(\lambda_1, \dots, \lambda_N)^{\{\underline{\alpha}'\}\{\underline{\beta}'\}}, \quad (B.1)$$

where $\underline{\alpha}' = (\alpha_0, \dots, \alpha_N)$ and $\underline{\beta}' = (\beta_0, \dots, \beta_N)$ with $\alpha_0 = \beta_0$ and $\alpha_N = \beta_N$. This gives immediately

$$\sum_{\alpha_{N+1}} A_N(\lambda_1, \dots, \lambda_N)[D_{N+1}(\lambda_1, \dots \lambda_N, \lambda_N + \lambda)]^{\underline{\alpha}\underline{\beta}} =$$

$$\qquad\qquad\qquad A_N(\lambda_1, \dots, \lambda_N)[D_N(\lambda_1, \dots \lambda_N)]^{\underline{\alpha}'\underline{\beta}'}. \quad (B.2)$$

Note that this fixes the spin $\gamma$ to be equal to $\alpha_1$ in the graphical representation above (or $\alpha_{N-1}$ for general $N$). Therefore we have obtained the left-hand side of the functional equation (28).

For the second step we sum over $\alpha = \alpha_N = \beta_N$ and fix the spin $\gamma$ to be equal to $\beta_{N-1}$. For $N = L = 2$ this becomes (thick dotted lines indicate where the constraint $\delta_{\gamma\beta_1}$ is used)

$$\delta_{\gamma\beta_1}\sum_{\alpha_2} A_2(\lambda_1, \lambda_2)[D_3(\lambda_1, \lambda_2, \lambda_2 + \lambda)]^{\{\underline{\alpha}\}\{\underline{\beta}\}} =$$

$$= \frac{\delta_{\alpha_0\beta_0}}{\rho(\lambda_1 - \lambda_2)\rho(\lambda_2 - \lambda_1)\Lambda(\lambda_1)\Lambda(\lambda_2)\Lambda(\lambda_2 + \lambda)} \times$$

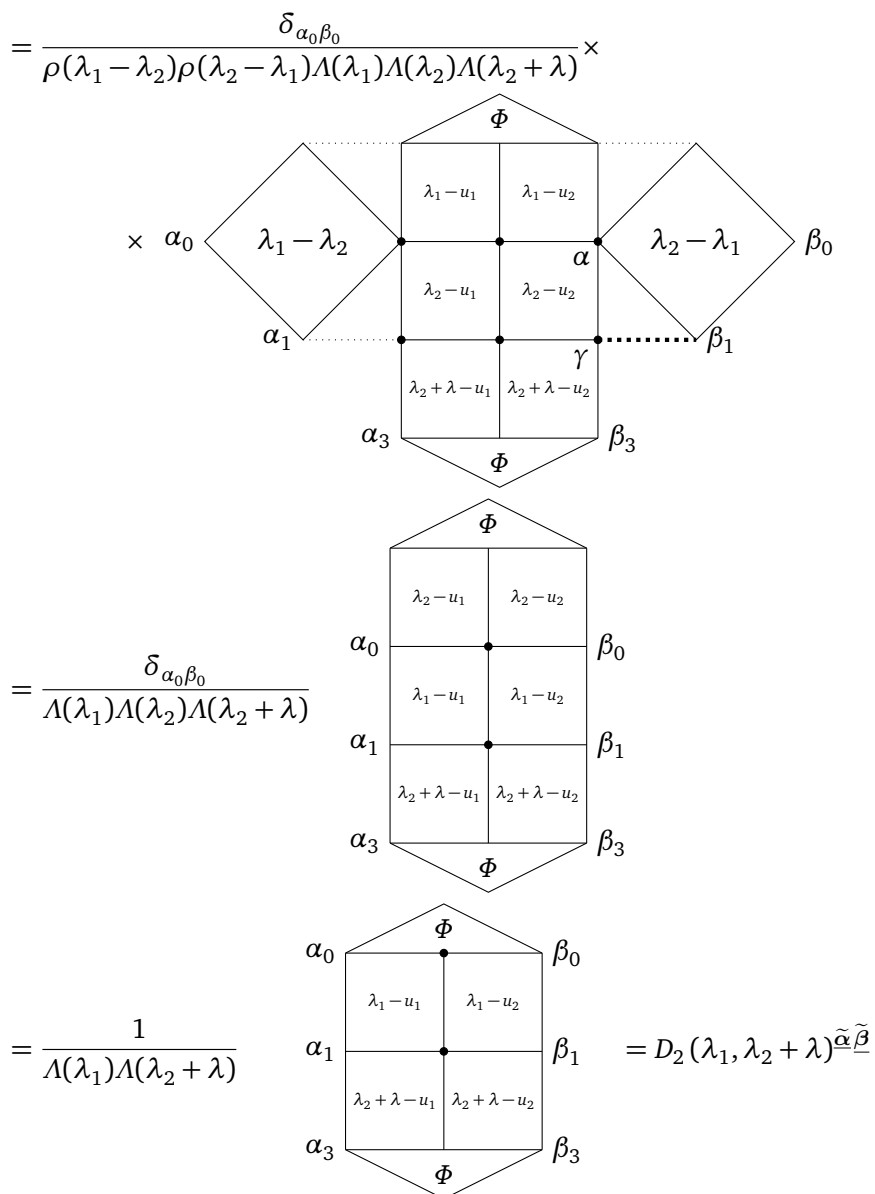

with $\widetilde{\underline{\alpha}} = (\alpha_0, \alpha_1, \alpha_3)$ and $\widetilde{\underline{\beta}}$ accordingly. Here we have used the initial and crossing conditions to evaluate the Boltzmann weight at $\lambda$, the Yang-Baxter equation to pull the rotated weight from the right to the left and finally $\langle \Phi | T_{\alpha_0 \alpha_0}(\lambda_2) = \Lambda(\lambda_2) \langle \Phi | P_{\alpha_0 \alpha_0}$ with the projection operator $P_{\alpha\alpha} : \mathcal{H}_{\text{per}} \to \mathcal{H}_{\alpha\alpha}$. For general $N > 2$ these operations have to be iterated to move the row $T(\lambda_N)$ to the top yielding the right-hand side of Eq. (28).

The final step consists of showing that for the special choice of $\lambda_N$ the two operations shown above yield the same result. Combining the unitarity condition (14) and crossing symmetry

(15) it follows immediately that

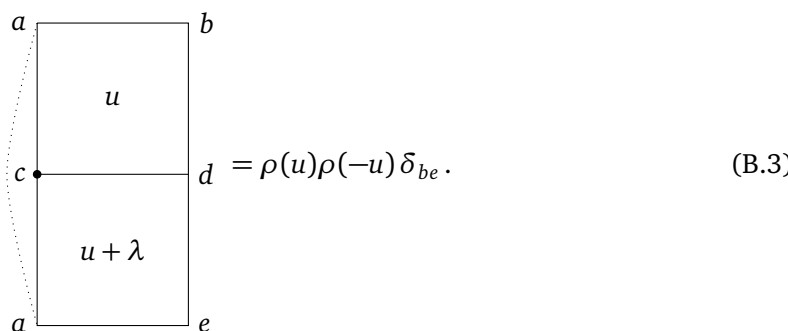

$$= \rho(u)\rho(-u)\,\delta_{be}. \tag{B.3}$$

This relation can be iterated which was used to find inversion relations for the transfer matrices of inhomogeneous face models [49]. Here it is the key ingredient to complete the proof. To this end we focus on the last two lines of $A_N[D_{N+1}](\lambda_1,\ldots,u_i,u_i+\lambda)$. The scalar prefactors appearing in (27) and in the following operations are suppressed as they do not depend on the spins in the auxiliary spaces and are irrelevant for the proof. Now we use the initial condition in the $i$-th column and (B.3) in the following ones until the rightmost line of spins is reached:

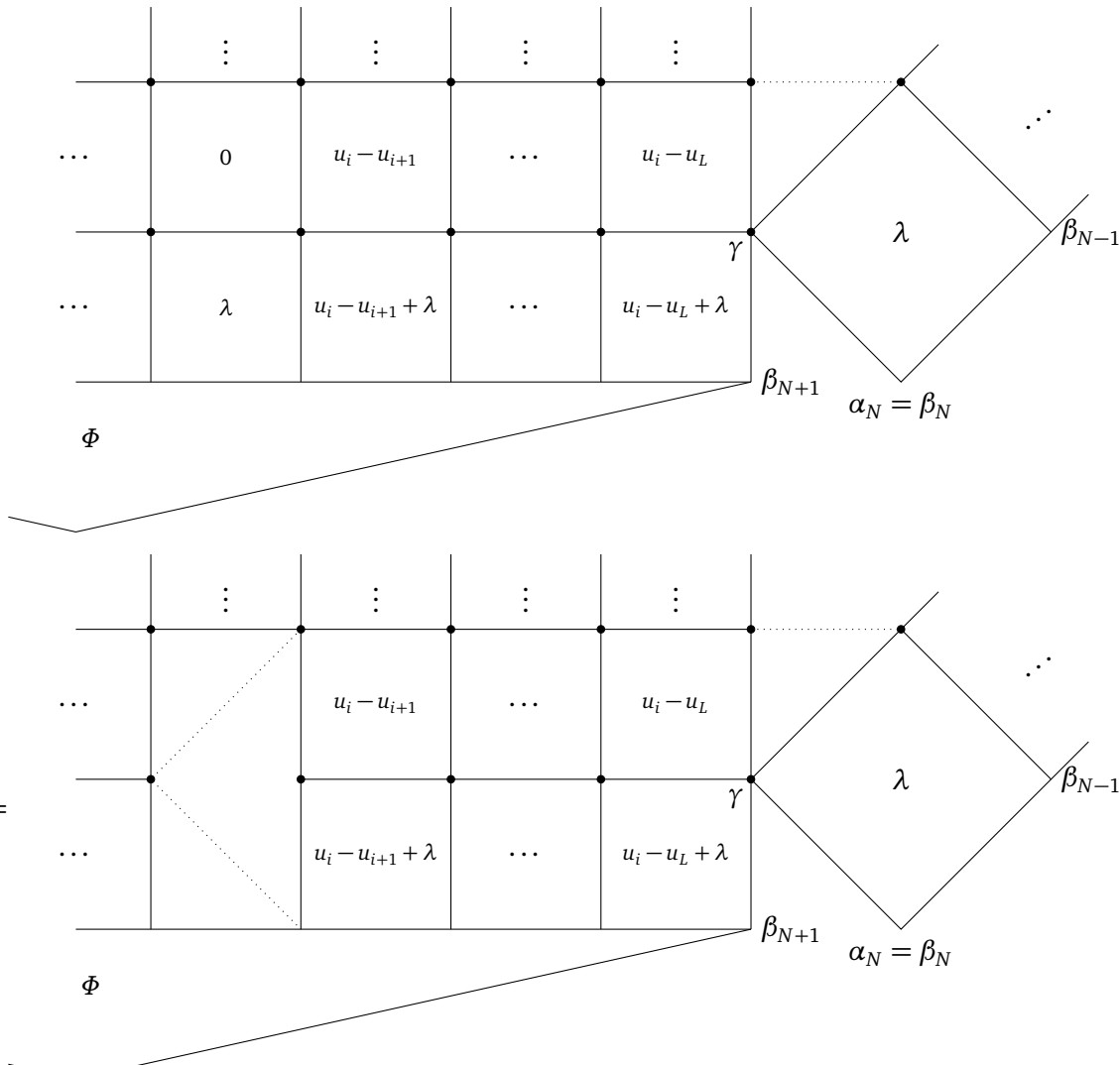

The two large diagrams here are lattice/graphical representations of the Boltzmann weight identities described in the proof. They involve spins $\beta_N$, $\beta_{N+1}$, $\beta_{N-1}$, $\beta_{N-2}$, $\alpha_N = \beta_N$, $\gamma$, $\Phi$, with spectral parameters $u_i - u_L$, $u_i - u_L + \lambda$, and $\lambda_N - \lambda_{N-1}$.

Using the initial condition for the rotated weight with spectral parameter $\lambda$ we obtain that $\beta_N = \beta_{N+1}$. By definition of the operator $A_N$ and periodic boundary conditions in quantum space $\mathcal{H}_{\text{per}}$ we also have $\alpha_N = \alpha_{N+1}$. The spin $\gamma$ is not connected to one of the Boltzmann weights any more and therefore the partial traces considered above yield the same result. This proves the theorem.

Note that the restriction of $\lambda_N \in \{u_i\}$ in the functional equation (28) can be dropped for matrix elements where $\alpha_{N-1}$ is a leaf node of the adjacency graph $\mathfrak{G}$: in this case all neighbouring spins are necessarily equal and therefore the lowest two rows can be removed using (B.3) for arbitrary values of $\lambda_N$.

Finally one should remark that, depending on the definition of the Boltzmann weigths for a particular model, the crossing relation may be modified by height dependent gauge factors, see e.g. (45) for the RSOS model. While these factors cancel in calculations where periodic boundary conditions can be imposed they have to be taken care of in the functional equation (28) – either by rescaling the operators $A$ and $D$ or by adding constant (i.e. not spectral parameter dependent) prefactors.

# C  Factorization of $D_3$ for the $r = 5$ RSOS model

Similar as in the $r = 4$ case, Eq. (63), the matrix elements of the three-site density operator $D_3(\lambda_1, \lambda_2, \lambda_3)$ in transfer matrix eigenstates from the $j = 0$ sector of the $r = 5$ RSOS model can be decomposed in terms factorizing into the two-point function $f(\lambda_i, \lambda_j)$, $1 \leq i < j \leq 3$, as defined in (73) and a set of structure functions $f_{i,j}(\lambda_1, \lambda_2, \lambda_3)$, i.e.

$$f_0 + f_{1,2}(\lambda_1, \lambda_2, \lambda_3) f(\lambda_1, \lambda_2) + f_{2,3}(\lambda_1, \lambda_2, \lambda_3) f(\lambda_2, \lambda_3) + f_{1,3}(\lambda_1, \lambda_2, \lambda_3) f(\lambda_1, \lambda_3).$$

Furthermore we find that all structure functions can be written as

$$f_{1,2}(\lambda_1, \lambda_2, \lambda_3) = \frac{1}{4} \left( f_{1,2}^1 + f_{1,2}^2 \cot(\lambda_{13}) + f_{1,2}^3 \cot(\lambda_{23}) + f_{1,2}^4 \cot(\lambda_{13}) \cot(\lambda_{23}) \right)$$

$$f_{1,3}(\lambda_1, \lambda_2, \lambda_3) = \frac{1}{4} \left( f_{1,3}^1 + f_{1,3}^2 \cot(\lambda_{12}) + f_{1,3}^3 \cot(\lambda_{23}) + f_{1,3}^4 \cot(\lambda_{12}) \cot(\lambda_{23}) \right) \qquad \text{(C.1)}$$

$$f_{2,3}(\lambda_1, \lambda_2, \lambda_3) = \frac{1}{4} \left( f_{2,3}^1 + f_{2,3}^2 \cot(\lambda_{12}) + f_{2,3}^3 \cot(\lambda_{13}) + f_{2,3}^4 \cot(\lambda_{12}) \cot(\lambda_{13}) \right),$$

where $f_0$ and $\{f_{i,j}^1, f_{i,j}^2, f_{i,j}^3, f_{i,j}^4\}$ are constants depending on the considered matrix element. Hence, we can uniquely describe any matrix element by in total 13 constants. In Table 1 we list these constants for the non-zero matrix elements $\langle \underline{\alpha} | D_3(\lambda_1, \lambda_2, \lambda_3) | \underline{\beta} \rangle$ in the sector with odd $\alpha_0$. All other matrix elements can be obtained by using reflection symmetry.

Table 1: Coefficients of the structure functions (C.1) appearing in the three site density operator for the $r = 5$ RSOS model.

| $\underline{\alpha}$ | $\underline{\beta}$ | $f_0$ | $\begin{array}{c} f^1_{1,2} \\ f^1_{1,3} \\ f^1_{2,3} \end{array}$ | $\begin{array}{c} f^2_{1,2} \\ f^2_{1,3} \\ f^2_{2,3} \end{array}$ | $\begin{array}{c} f^3_{1,2} \\ f^3_{1,3} \\ f^3_{2,3} \end{array}$ | $\begin{array}{c} f^4_{1,2} \\ f^4_{1,3} \\ f^4_{2,3} \end{array}$ |
|---|---|---|---|---|---|---|
| (1212) | (1212) | $\frac{1}{2}-\frac{1}{\sqrt5}$ | $\begin{array}{c} 4 \\ 0 \\ 0 \end{array}$ | $\begin{array}{c} 0 \\ 0 \\ 0 \end{array}$ | $\begin{array}{c} 0 \\ 0 \\ 0 \end{array}$ | $\begin{array}{c} 0 \\ 0 \\ 0 \end{array}$ |
| (1212) | (1232) | $0$ | $\begin{array}{c} -\sqrt{5\left(\sqrt5-2\right)} \\ -\sqrt{\sqrt5+2} \\ \sqrt{\sqrt5+2} \end{array}$ | $\begin{array}{c} \sqrt[4]{5} \\ -\sqrt[4]{5} \\ \sqrt[4]{5} \end{array}$ | $\begin{array}{c} -\sqrt[4]{5} \\ \sqrt[4]{5} \\ -\sqrt[4]{5} \end{array}$ | $\begin{array}{c} -\sqrt{5\left(\sqrt5-2\right)} \\ -\sqrt{5\left(\sqrt5-2\right)} \\ \sqrt{5\left(\sqrt5-2\right)} \end{array}$ |
| (1232) | (1212) | $0$ | $\begin{array}{c} -\sqrt{5\left(\sqrt5-2\right)} \\ -\sqrt{\sqrt5+2} \\ \sqrt{\sqrt5+2} \end{array}$ | $\begin{array}{c} -\sqrt[4]{5} \\ \sqrt[4]{5} \\ -\sqrt[4]{5} \end{array}$ | $\begin{array}{c} \sqrt[4]{5} \\ -\sqrt[4]{5} \\ \sqrt[4]{5} \end{array}$ | $\begin{array}{c} -\sqrt{5\left(\sqrt5-2\right)} \\ \sqrt{5\left(\sqrt5-2\right)} \\ -\sqrt{5\left(\sqrt5-2\right)} \end{array}$ |
| (1232) | (1232) | $\frac{1}{2}-\frac{1}{\sqrt5}$ | $\begin{array}{c} \sqrt5-3 \\ 1+\sqrt5 \\ 1+\sqrt5 \end{array}$ | $\begin{array}{c} 0 \\ 0 \\ 0 \end{array}$ | $\begin{array}{c} 0 \\ 0 \\ 0 \end{array}$ | $\begin{array}{c} 3\sqrt5-5 \\ -(3\sqrt5-5) \\ 3\sqrt5-5 \end{array}$ |
| (1234) | (1234) | $\frac{7}{4\sqrt5}-\frac{3}{4}$ | $\begin{array}{c} -(\sqrt5+1) \\ -(\sqrt5+1) \\ -(\sqrt5+1) \end{array}$ | $\begin{array}{c} 0 \\ 0 \\ 0 \end{array}$ | $\begin{array}{c} 0 \\ 0 \\ 0 \end{array}$ | $\begin{array}{c} (3\sqrt5-5) \\ (3\sqrt5-5) \\ -(3\sqrt5-5) \end{array}$ |

| $\underline{\alpha}$ | $\underline{\beta}$ | $f_0$ | $\begin{matrix}f^1_{1,2}\\f^1_{1,3}\\f^1_{2,3}\end{matrix}$ | $\begin{matrix}f^2_{1,2}\\f^2_{1,3}\\f^2_{2,3}\end{matrix}$ | $\begin{matrix}f^3_{1,2}\\f^3_{1,3}\\f^3_{2,3}\end{matrix}$ | $\begin{matrix}f^4_{1,2}\\f^4_{1,3}\\f^4_{2,3}\end{matrix}$ |
|---|---|---|---|---|---|---|
| (3212) | (3212) | $\dfrac{3}{4\sqrt5}-\dfrac14$ | $\begin{matrix}-4\\0\\0\end{matrix}$ | $\begin{matrix}0\\0\\0\end{matrix}$ | $\begin{matrix}0\\0\\0\end{matrix}$ | $\begin{matrix}0\\0\\0\end{matrix}$ |
| (3212) | (3232) | $0$ | $\begin{matrix}\sqrt{5\left(\sqrt5-2\right)}\\\sqrt{\sqrt5+2}\\3\sqrt{\sqrt5+2}\end{matrix}$ | $\begin{matrix}-\sqrt[4]5\\\sqrt[4]5\\-\sqrt[4]5\end{matrix}$ | $\begin{matrix}\sqrt[4]5\\-\sqrt[4]5\\\sqrt[4]5\end{matrix}$ | $\begin{matrix}\sqrt{5\left(\sqrt5-2\right)}\\-\sqrt{5\left(\sqrt5-2\right)}\\5\left(\sqrt5-2\right)\end{matrix}$ |
| (3212) | (3432) | $0$ | $\begin{matrix}-\tfrac12\left(\sqrt5+5\right)\\\tfrac12\left(-3\sqrt5-7\right)\\-\tfrac12\left(\sqrt5+5\right)\end{matrix}$ | $\begin{matrix}-\sqrt{\tfrac12\left(5-\sqrt5\right)}\\\sqrt{\tfrac12\left(5-\sqrt5\right)}\\-\sqrt{\tfrac12\left(5-\sqrt5\right)}\end{matrix}$ | $\begin{matrix}\sqrt{\tfrac12\left(5-\sqrt5\right)}\\-\sqrt{\tfrac12\left(5-\sqrt5\right)}\\\sqrt{\tfrac12\left(5-\sqrt5\right)}\end{matrix}$ | $\begin{matrix}-\tfrac12\left(\sqrt5+5\right)\\\tfrac12\left(\sqrt5+5\right)\\-\tfrac12\left(\sqrt5+5\right)\end{matrix}$ |
| (3232) | (3212) | $0$ | $\begin{matrix}\sqrt{5\left(\sqrt5-2\right)}\\\sqrt{\sqrt5+2}\\3\sqrt{\sqrt5+2}\end{matrix}$ | $\begin{matrix}\sqrt[4]5\\-\sqrt[4]5\\\sqrt[4]5\end{matrix}$ | $\begin{matrix}-\sqrt[4]5\\\sqrt[4]5\\-\sqrt[4]5\end{matrix}$ | $\begin{matrix}\sqrt{5\left(\sqrt5-2\right)}\\-\sqrt{5\left(\sqrt5-2\right)}\\5\left(\sqrt5-2\right)\end{matrix}$ |
| (3232) | (3232) | $\dfrac{3}{4\sqrt5}-\dfrac14$ | $\begin{matrix}3-\sqrt5\\-\sqrt5-1\\3-\sqrt5\end{matrix}$ | $\begin{matrix}0\\0\\0\end{matrix}$ | $\begin{matrix}0\\0\\0\end{matrix}$ | $\begin{matrix}5-3\sqrt5\\3\sqrt5-5\\5-3\sqrt5\end{matrix}$ |
| (3232) | (3432) | $0$ | $\begin{matrix}3\sqrt{\sqrt5+2}\\\sqrt{\sqrt5+2}\\\sqrt{5\left(\sqrt5-2\right)}\\-\tfrac12\left(\sqrt5+5\right)\end{matrix}$ | $\begin{matrix}-\sqrt[4]5\\\sqrt[4]5\\-\sqrt[4]5\\\sqrt{\tfrac12\left(5-\sqrt5\right)}\end{matrix}$ | $\begin{matrix}\sqrt[4]5\\-\sqrt[4]5\\\sqrt[4]5\\-\sqrt{\tfrac12\left(5-\sqrt5\right)}\end{matrix}$ | $\begin{matrix}\sqrt{5\left(\sqrt5-2\right)}\\-\sqrt{5\left(\sqrt5-2\right)}\\5\left(\sqrt5-2\right)\\-\tfrac12\left(\sqrt5+5\right)\end{matrix}$ |

| $\underline{\alpha}$ | $\underline{\beta}$ | $f_0$ | $\begin{matrix}f^1_{1,2}\\f^1_{1,3}\\f^1_{2,3}\end{matrix}$ | $\begin{matrix}f^2_{1,2}\\f^2_{1,3}\\f^2_{2,3}\end{matrix}$ | $\begin{matrix}f^3_{1,2}\\f^3_{1,3}\\f^3_{2,3}\end{matrix}$ | $\begin{matrix}f^4_{1,2}\\f^4_{1,3}\\f^4_{2,3}\end{matrix}$ |
|---|---|---|---|---|---|---|
| (3432) | (3212) | 0 | $\frac{1}{2}(-3\sqrt{5}-7)$ <br> $-\frac{1}{2}(\sqrt{5}+5)$ | $-\sqrt{\frac{1}{2}(5-\sqrt{5})}$ <br> $\sqrt{\frac{1}{2}(5-\sqrt{5})}$ | $\sqrt{\frac{1}{2}(5-\sqrt{5})}$ <br> $-\sqrt{\frac{1}{2}(5-\sqrt{5})}$ | $\frac{1}{2}(\sqrt{5}+5)$ <br> $-\frac{1}{2}(\sqrt{5}+5)$ |
| (3432) | (3232) | 0 | $3\sqrt{\sqrt{5}+2}$ <br> $\sqrt{\sqrt{5}+2}$ <br> $\sqrt{5(\sqrt{5}-2)}$ | $\sqrt[4]{5}$ <br> $-\sqrt[4]{5}$ <br> $\sqrt[4]{5}$ | $-\sqrt[4]{5}$ <br> $\sqrt[4]{5}$ <br> $-\sqrt[4]{5}$ | $\sqrt{5(\sqrt{5}-2)}$ <br> $-\sqrt{5(\sqrt{5}-2)}$ <br> $\sqrt{5(\sqrt{5}-2)}$ |
| (3432) | (3432) | $\frac{3}{4\sqrt{5}}-\frac{1}{4}$ | 0 <br> 0 <br> $-4$ | 0 <br> 0 <br> 0 | 0 <br> 0 <br> 0 | 0 <br> 0 <br> 0 |

| $\underline{\alpha}$ | $\underline{\beta}$ | $f_0$ | $\begin{matrix}f^1_{1,2}\\f^1_{1,3}\\f^1_{2,3}\end{matrix}$ | $\begin{matrix}f^2_{1,2}\\f^2_{1,3}\\f^2_{2,3}\end{matrix}$ | $\begin{matrix}f^3_{1,2}\\f^3_{1,3}\\f^3_{2,3}\end{matrix}$ | $\begin{matrix}f^4_{1,2}\\f^4_{1,3}\\f^4_{2,3}\end{matrix}$ |
|---|---|---|---|---|---|---|
| (3234) | (3234) | $\frac{1}{2}-\frac{1}{\sqrt{5}}$ | $\sqrt{5}+1$ <br> $\sqrt{5}+1$ <br> $\sqrt{5}-3$ | $0$ <br> $0$ <br> $0$ | $0$ <br> $0$ <br> $0$ | $3\sqrt{5}-5$ <br> $-(3\sqrt{5}-5)$ <br> $3\sqrt{5}-5$ |
| (3234) | (3434) | $0$ | $\sqrt{\sqrt{5}+2}$ <br> $-\sqrt{\sqrt{5}+2}$ <br> $-\sqrt{5\left(\sqrt{5}-2\right)}$ | $\sqrt[4]{5}$ <br> $-\sqrt[4]{5}$ <br> $\sqrt[4]{5}$ | $-\sqrt[4]{5}$ <br> $\sqrt[4]{5}$ <br> $-\sqrt[4]{5}$ | $-\sqrt{5\left(\sqrt{5}-2\right)}$ <br> $\sqrt{5\left(\sqrt{5}-2\right)}$ <br> $-\sqrt{5\left(\sqrt{5}-2\right)}$ |
| (3434) | (3234) | $0$ | $\sqrt{\sqrt{5}+2}$ <br> $-\sqrt{\sqrt{5}+2}$ <br> $-\sqrt{5\left(\sqrt{5}-2\right)}$ | $-\sqrt[4]{5}$ <br> $\sqrt[4]{5}$ <br> $-\sqrt[4]{5}$ | $\sqrt[4]{5}$ <br> $-\sqrt[4]{5}$ <br> $\sqrt[4]{5}$ | $-\sqrt{5\left(\sqrt{5}-2\right)}$ <br> $\sqrt{5\left(\sqrt{5}-2\right)}$ <br> $-\sqrt{5\left(\sqrt{5}-2\right)}$ |
| (3434) | (3434) | $\frac{1}{2}-\frac{1}{\sqrt{5}}$ | $0$ <br> $0$ <br> $4$ | $0$ <br> $0$ <br> $0$ | $0$ <br> $0$ <br> $0$ | $0$ <br> $0$ <br> $0$ |

# D  Non-abelian anyons

The sequence of fusion processes defining the fusion path basis (4) of states for an anyonic quantum chain can be reordered by so-called F-moves [32]. Up to some gauge freedom the latter are determined by the fusion rules (3) and the pentagon equation. Here we define the F-moves graphically

$$
\begin{array}{c}
\begin{array}{cc} b & c \end{array} \\
\underline{\phantom{a \quad e \quad d}} \\
\begin{array}{ccc} a & e & d \end{array}
\end{array}
= \sum_f \left( F_d^{abc} \right)_f^e \;\;
\begin{array}{c}
\begin{array}{cc} b & c \end{array} \\
f \\
\underline{\phantom{a \qquad\qquad d}} \\
\begin{array}{cc} a & d \end{array}
\end{array}
\; . \tag{D.1}
$$

For Fibonacci anyons the only non-trivial F-move is

$$
\left( F_\tau^{\tau\tau\tau} \right)_f^e = \begin{pmatrix} \phi^{-1} & \phi^{-1/2} \\ \phi^{-1/2} & -1/\phi \end{pmatrix}_{ef} , \tag{D.2}
$$

where $e, f \in \{1, \tau\}$ and $\phi \equiv \frac{1}{2}(1+\sqrt{5})$ is the golden ratio (see e.g. Ref. [28]. All others are 1 if they are allowed by the fusion rules and 0 else.

Local projection operators can be expressed in terms of the F-moves as

$$
P_i^{(\tau\tau\to\ell)} \equiv \sum_{a_{i-1}, a_i, a_i', a_{i+1}} \left[ \left( F_{a_{i+1}}^{a_{i-1}\tau\tau} \right)_\ell^{a_i'} \right]^* \left( F_{a_{i+1}}^{a_{i-1}\tau\tau} \right)_\ell^{a_i} |\ldots a_{i-1} a_i' a_{i+1} \ldots\rangle\langle\ldots a_{i-1} a_i a_{i+1} \ldots|. \tag{D.3}
$$

Though depending on the sites $i-1, i$ and $i+1$ they leave the first and the last invariant. Since the F-moves of the Fibonacci anyons are real valued, we can drop the complex conjugation. A straight forward generalization allows to express local three anyon projection operators as

$$
\langle\boldsymbol{a}| P_i^{(\tau\tau\to\ell)} |\boldsymbol{b}\rangle \equiv \left( \prod_{k\notin\{i,i+1\}} \delta_{a_k b_k} \right) \sum_x \left[ \left( F_{a_{i+1}}^{b_{i-1}\tau\tau} \right)_x^{a_i} \left( F_{b_{i+2}}^{b_{i-1}x\tau} \right)_\ell^{a_{i+1}} \right]^* \left( F_{b_{i+2}}^{b_{i-1}x\tau} \right)_\ell^{b_{i+1}} \left( F_{b_{i+1}}^{b_{i-1}\tau\tau} \right)_x^{b_i} . \tag{D.4}
$$

The anyons of an $su(2)_3$ theory can be labeled by generalized spins $j = 0, \frac{1}{2}, 1, \frac{3}{2}$. An automorphism of the corresponding fusion algebra allows to identify $j = 0, \frac{3}{2}$ with the trivial ($x = 1$) and $j = \frac{1}{2}, 1$ with the $\tau$-anyon of the Fibonacci chain. Starting from fusion path states $|x_0 x_1 \ldots x_L\rangle$ of Fibonacci anyons with $x_n \in \{1, \tau\}$ we obtain the Hilbert space of $r = 5$ RSOS model defined in Section 5.2 by mapping

$$
x_n \mapsto a_n \equiv \begin{cases} 1 & \text{for } x_n = 1, \quad n \text{ odd} \\ 2 & \text{for } x_n = \tau, \quad n \text{ even} \\ 3 & \text{for } x_n = \tau, \quad n \text{ odd} \\ 4 & \text{for } x_n = 1, \quad n \text{ even} \end{cases} . \tag{D.5}
$$

Note that this mapping gives only half of the basis states of the RSOS model, since $a_n$ will be even (odd) on the even (odd) sublattice. The other half is obtained by switching odd and even in (D.5). This also provides a mapping of the anyon Hamiltonian to an operator in the RSOS model. Similarly the projection operators $P_i^{(\tau\tau\to 1)}$ are mapped (up to a factor $\phi$) to local operators $e_i$ forming a representation of the Temperley-Lieb algebra [28, 50]

$$
\langle\boldsymbol{a}| e_i |\boldsymbol{b}\rangle = \delta_{a_{i-1} a_{i+1}} \sqrt{\frac{g_{a_i} g_{b_i}}{g_{a_{i-1}} g_{a_{i+1}}}} \prod_{k\neq i} \delta_{a_k b_k} , \tag{D.6}
$$

with gauge factors $g_x$ from Eq. (44). Comparing this with the RSOS Boltzmann-weights (43) we find:

$$\langle \boldsymbol{a}|e_i|\boldsymbol{b}\rangle = \left(\prod_{k\neq i}\delta_{a_k b_k}\right) W\begin{pmatrix} a_{i-1} & a_i \\ b_i & a_{i+1} \end{pmatrix}\lambda\right). \tag{D.7}$$

We will now express the Hamiltonian by means of the transfer matrix. Therefore, we observe that

$$W'\begin{pmatrix} a_{i-1} & a_i \\ b_i & a_{i+1} \end{pmatrix} 0\right) = \frac{1}{\sin\lambda} W\begin{pmatrix} a_{i-1} & a_i \\ b_i & a_{i+1} \end{pmatrix}\lambda\right) - \cot\lambda\,\delta_{b_i a_i}, \tag{D.8}$$

where $W'$ is the derivative with respect to the spectral parameter $u$. Furthermore, crossing symmetry and unitarity imply that $t^{-1}(0) = t(\lambda)$ for the homogeneous model (i.e. all inhomogeneities $u_i = 0$). In that case, both $t(0)$ and $t(\lambda)$ are shift operators due to the initial condition. Hence, the logarithmic derivative of the transfer matrix at $u = 0$ is the sum of local operators

$$\langle \boldsymbol{a}|t^{-1}(0)t'(0)|\boldsymbol{b}\rangle = \sum_i \left( \frac{1}{\sin\lambda} \cdots \right. \left. \cdots - \cot\lambda \prod_{k=0}^{L}\delta_{a_k b_k} \right)$$

$$= \sum_i \langle \boldsymbol{a}|\frac{e_i}{\sin\lambda} - \cot\lambda\,\mathbb{1}|\boldsymbol{b}\rangle \tag{D.9}$$

or

$$\sum_i e_i = \sin\lambda\, t^{-1}(0)t'(0) + L\cos\lambda\mathbb{1}. \tag{D.10}$$

Hence the anyonic Hamiltonian (81)

$$H = J\sum_i P_i^{(\tau\tau\to 1)} \mapsto \frac{J}{\phi}\sum_i e_i \tag{D.11}$$

is a member of the commuting family of operators generated by the transfer matrix (9) of the $r = 5$ RSOS model.

Likewise we can also map the 3-anyon projector (D.4) to an operator acting on the $r = 5$ RSOS model Hilbert space.

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
