# Peer review of "Density matrices in integrable face models"

_SciPost Physics, doi:SciPost Phys. 11, 057 (2021)_

## Round 1 · Referee Report · Anonymous (Referee 1) · 2021-5-17

Report

In this work, the authors study integrable face models (sometimes called IRF or SOS models) with inhomogeneity parameters and derive some discrete functional equations for their reduced density matrices. These discrete equations may be considered as some finite-size analog of the qKZ equations. Similar equations have already been obtained in the literature for temperature correlation functions in the Heisenberg spin chain. The derivation of these equations rely on basic algebraic properties satisfied by the Boltzmann weights of model, such as the Yang-Baxter equation, the unitarity condition, the crossing symmetry and the initial condition. In a last part of the paper, the functional equation is used to explicitly compute the density matrix elements on 2 or 3 sites in some particular examples: the critical CSOS model, the critical RSOS model, and a model of Fibonacci anyons.

I find this work interesting and a priori suitable for publications in SciPost Physics. However, I have several remarks or questions.

1) References to previous works in not always accurate and sometimes confused in the introduction:

  • On page 2, it is written: “while the corresponding modified (dynamical) Yang-Baxter algebra allows for a solution of the spectral problem by means of an algebraic Bethe ansatz [11] or in the framework of Sklyanin’s Separation of Variables [12, 13] this modification does, however, pose a problem for the computation of physical quantities such as correlation functions. One reason is the lack of a one-to-one mapping between height configurations in the SOS formulation and spin configurations in the dynamical six-vertex model which may lead to boundary contributions to certain expectation values even in the thermodynamic limit [14]. "

I find this sentence extremely confusing. Correlation functions for SOS models where for instance computed in:

[R1] S. Lukyanov and Y. Pugai, Multi-point local height probabilities in the integrable RSOS model, Nucl. Phys. B 473 (1996), 631–658. 

[R2] D. Levy-Bencheton and V. Terras, Multi-point local height probabilities of the CSOS model within the algebraic Bethe ansatz framework, J. Stat. Mech. (2014) P04014.

I don’t see in these two works any problem with a "lack of a one-to-one mapping between height configurations in the SOS formulation and spin configurations in the dynamical six-vertex model". In fact, the two formulations are in my understanding completely equivalent. Maybe the authors refer to the correspondence introduced by Baxter with the eight-vertex model, which is in fact not a one-to-one mapping (and is indeed a problem for the computation of correlation functions of the eight-vertex model itself, but not for those of the SOS model)? Or maybe they refer to the fact that the particular restriction of generic SOS models into RSOS model, for which the height are chosen in a finite set, may lead to some difficulties in the writing of some ABA solution of the model? But even in that case other approaches to correlation functions are possible, as in [R1], and the difficulty is not particularly linked to a rewriting or not of the algebra in the Felder and Varchenko [11] (i.e. dynamical) way. Hence, this sentence should be modified and clarified, and the two above works [R1] and [R2] about the computation of correlation functions in such models should be added here. One have indeed the wrong impression when reading the introduction that nothing has been done so far concerning the computation of correlation functions of SOS models.

By the way, in the aforementioned sentence, whether [11] refers to the ABA solution of the SOS models relying on the dynamical Yang-Baxter algebra, [12,13] do not refer at all to SOS models! The Separation of Variables solution of such models have been done in

[R3] G. Niccoli, An antiperiodic dynamical six-vertex model: I. Complete spectrum by SOV, matrix elements of the identity on separate states and connections to the periodic eight-vertex model, J. Phys. A: Math. Theor. 46 (2013), 075003

and in Reference [19] of the present paper. The two references [R3] and [19] should be cited here.

  • In the next sentence, it is written “the inverse problem relating local observables to elements of the Yang-Baxter algebra [16,17], have been obtained only for the particular cases of the cyclic (CSOS) and the antiperiodic SOS models such that multi-point height probabilities could be calculated [18, 19]". In this sentence is mixed at the same level two different things: a particular restriction of the model (CSOS), and a particular choice of boundary conditions for the general SOS model. However, the solution of the inverse problem of [18] (formulated there for periodic boundary conditions) is not particularly specific to CSOS models since it uses only basic properties of the R-matrix of the model which are strictly equivalent to (2.12)-(2.15) of the present paper (as far as I understand, the range of values of the heights play no role there). As a matter of fact, it was directly generalized in [19] to any twisted boundary conditions (and not only antiperiodic), where there a completely general SOS model is considered. So the solution of the inverse problem obtained in [18,19] seems to me not less general as the one obtained in the present paper. This sentence should therefore be modified.

  • The next sentence, "we address some of the issues appearing in the calculation of correlation functions in generic face models in particular on finite lattices while avoiding the use of the correspondence to a dynamical vertex model", is also confusing. As already mentioned above, I do not understand what the authors mean by a “correspondence” with a dynamical vertex model at this stage. Do they speak about the vertex-IRF correspondence? But this usually means the correspondence introduced by Baxter to study the eight-vertex model from SOS, which is of course not relevant here. Or do they mean the reformulation of the Boltzmann weights of the SOS model as the elements of some R-matrix solution of a dynamical Yang-Baxter equation, as in Felder’s works? But this is purely formal to underline the algebra behind the model, and cannot be called “correspondence”: at the level of the present paper it is just a question of choice of notations. Once again, the authors should clarify what they mean here and modify the sentence accordingly.

2) Notations and definitions could sometimes be improved:

  • On equation (2.1), what does it mean to have a definition with “may be” and “may not be”? In a definition, one should use “are” or “are not”.

  • What does it mean to “read from top-left to bottom-right” on a graphic like at the top page 4? Please clarify.

  • There seems to be a mixing of notations where indices or subscripts represent the fact that the operator is “sandwiched” between states, as in (2.7), or where the subscripts or indices seem to represent only the corner values, as in (2.10). I find it not completely clear. Can the authors be more explicit when defining (2.10)?

  • Many crucial formulas are written only graphically. I find this not completely clear, especially when appears “turned squares” without explanations, considering that the identification of the upper left corner is important for the value of the Boltzmann weight. Although this type of representations is quite standard in the literature, I think it is important, for the article to be self-contained and easier to read, to have clear definitions without need to refer to previous papers to understand them. Can the authors write, for clarity, the explicit analytical form of the formulas (2.12), (2.13), (2.14), (2.15) and (4.1) in addition to the graphical one?

3) About the solution of the ‘inverse problem’ on p.7-8:

  • As mentioned above, the solution of the inverse problem of [18] seems to me as general as the one proposed in this section. But probably to consider the local operator (2.11) one has to combine the results of [18] with the local weights considered in [R2]. Maybe the authors could comment about this and on how their result could be related to the ones of [18]-[R2]? They should of course also modify the sentence “A first step towards the solution of the inverse problem for face models has been achieved in Ref. [18] for the cyclic solid-on-solid (CSOS) model which can be related to the eight-vertex model by considering a dynamical version of the Yang-Baxter equation [11, 26, 27]" accordingly.

  • Would it be possible to have a general proof of the result, with details for L>2 to see explicitly how the dressing of transfer matrices appear in the result?

4) About the applications of the functional equation:

  • Is there a reason for which the authors restrict their study to critical models? Is the functional equation too complicated to solve in the general case where the Boltzmann weights explicitly depend on the height variable?

  • The discrete functional equation obtained in [29] in temperature XXZ case was used there to show that the solution could be stated in terms of a single nearest neighbor correlator ω. The authors of the present paper observe a similar behavior in some topological sector of the RSOS model. However, in the CSOS case, they limit their study to the case N=2 and do not discuss this question. Is it because this property does not hold in that case? It would be nice that the authors comment about this since they conclude the paper by the sentence “this may well allow to shed some light on the question whether the factorization of correlation functions is a general property of integrable models which extends beyond RSOS models and spin-1/2 chains".

Requested changes

See report.

---

## Round 1 · Referee Report · Anonymous (Referee 2) · 2021-5-23

Strengths

1.) Probably the first paper which touches the factorization of static correlation functions of the face models, so the paper may be opening up a new and interesting direction

Weaknesses

1.) Presentation of main results could be improved
2.) Discussion of the context could be improved

Report

The authors launch a study of reduced density matrices of integrable face models. They express the reduced density matrices of the inhomogeneous models as normalized expectation values of products of monodromy operators, mimicking in a way the corresponding construction for the vertex models. They show that the inhomogenous reduced density matrices satisfy a discrete version of the rqKZ equation obtained in [35,29]. This equation is then used to study the reduced density matrices on two and three lattice sites for SOS and RSOS models and also for a related model of non-Abelian anyons. As concrete results the authors obtain the two-site reduced density matrix of the SOS model for a vacuum state and relations for some of the three-site reduced density matrices of two RSOS models (r = 4, 5) with certain elements of the corresponding two-site density matrix. The latter is then also upgraded to a special anyon model.

To my best knowlegde this is the first observation of a factorization property of static correlation functions, that was known for the XXZ chain (or the associated six-vertex model), in a face model. This observation is quite interesting as there have been speculations whether or not such a factorization would be a general property of Yang-Baxter integrable models. For this reason I recommend the publication of the paper in SciPost.

Being the second referee I have the privilege of having read the first referee report. Like the first referee I am not entirely happy with the presentation of the results and with the way the authors try to make connection with the existing literature. As I read the first report I will refrain from futher commenting on theorem 1. I also agree on his/her remarks on the graphical notation.

In addition, I would like to comment on the factorization of the correlation functions of the XXZ model and, in passing, suggest to be more precise in certain statements and in making reference to previous work.

In the first paragraph of the introduction the authors write "$\dots$ a remarkable property of the reduced density matrices has been found [6,7]: correlation functions of spins on N consequtive sites can be expressed as sums of terms factorizing into products of nearest-neighbour (two-point) functions of the generalized model. Their coefficients are recursively defined elementary functions of N spectral parameters and do not depend on model parameters such as the system size or choice of inhomogenities." Here two things are mixed. In fact the factorization was explained in [6,7]. But these papers are based on the qKZ (or rqKZ) equation which is only valid for the infinite chain. Those times most of the protagonists in the field believed that factorization was a "property of the vacuum". Factorization at finite temperature (or finite length) was first observed in the study of the three-site reduced density matrix of the XXX chain in the works

[R1] Factorization of multiple integrals representing the density matrix of a finite segment of the Heisenberg spin chain, H. E. Boos et al., J. Stat. Mech. 0604: P04001, 2006

[R2] Density matrices for finite segments of Heisenberg chains of arbitrary length, J. Damerau et al., J. Phys. A 40 (2007) 4439

Those days these works were a surprise to many people. The factorization in the most general case of generalized density matrices of the XXZ chain was then proven in the works

[R3] Hidden Grassmann Structure in the XXZ Model II: Creation Operators, H. Boos et al., Commun. Math. Phys. 286: 875, 2009

[R4] Hidden Grassmann Structure in the XXZ Model III: Introducing Matsubara direction, M. Jimbo, T. Miwa and F. Smirnov, J. Phys. A 42: 304018, 2009

The approach based on "discrete functional equation", [35,29] came only later and offered an alternative proof of the factorization. I think it would be rather helpful to the readers if the original works would be mentioned and if it would be explained which are the advantages and disadvantages of the different methods.

In my understanding the situation is as follows. As compared to the Fermionic basis approach the approach using the discrete functional equations is much easier to generalize. One has to be careful though, as the functional equation approach of [35,29] requires, in my understanding, several more steps which are missing in the present paper. In [29] it was shown, that the discrete functional equations of rqKZ type when supplied with an asymptotic reduction, obtained by sending one of the spectral parameters to infinity, have a unique solution. Then the solution of the usual rqKZ equation [7] was plugged in and was shown to solve the functional equation und the asymptotic reduction. This provided an alternative proof of the factorization. However, this proof was not constructive. It required the preknowledge of the solution.

In the present work the authors claim in the first paragraph of the conclusions that the "discrete functional equation" [$\dots$] "determine the functional dependence of $D_N$ on these spectral parameter. Given the analytical properties inherited from the Boltzmann weights of the model considered these equations have a unique solution for any transfer matrix eigenstate." I am not sure what they want to say here. As far as I understand it, there is no proof of this statement in the paper and no refererence. Is this a conjecture? I think the statement cannot be true without further requirements like asymptotic conditions. A proof of such a statement for the Heisenberg model is given in [29]. Do the authors claim that a similar proof holds for the face models? They should be more specific here.

As I said above, in my understanding the uniqueness of the solutions of the discrete functional equations if supplied with appropriate asymptotic conditions is essential for the proof of the factorization of the correlation functions in [29]. Another ingredient of the proof is the Ansatz, taken from [7], for the factorized form of the reduced density matrix. This includes, in particular, knowledge of the number of unknown functions. The authors observe that `in certain topological sectors' the answer requires a single function. How many are there in general? Is it known or is there a conjecture?

So far, in case of the XXZ model, an advantage of the Fermionic basis approach over the approach using discrete functional equations is that it characterizes these unknown functions, which are one-point and two-point functions that constitute the beginning of the recursion. In the paper this concerns the function $f(\lambda, \mu)$. In the Fermionic basis approach there are two functions $\rho$ and $\omega$ which were characterized by means of integral formula and difference equations in [R4] and in terms of solutions of non-linear integral equations in the work

[R5] On the physical part of the factorized correlation functions of the XXZ chain, H. Boos and F. Göhmann, J. Phys. A 42: 315001, 2009

Equation (67) of [R5] shows that the functions $\omega$, expressed by means of solutions of nonlinear integral equations satisfies a basic discrete functional equation. In the approach of [29] this was used as an input. Such an input as well as the general form of the solution, taking from [7] in case of the XXX chain, seems to be still missing in the context of the face models. I think it would be helpful if the authors could be more precise here. They do obtain the function $g(\lambda, \mu)$ in section 5.1 for a very special case of the density matrix build from a vacuum state. It would be helpful if the authors could comment on the possibilities to calculate e.g. the function $f(\lambda, \mu)$ in (5.2).

What is also a pity for the curious readers is that the authors say in the introduction that they "propose an algorithm for the efficient computation of the structure coefficient", but this algorithm is only vaguely described in the body of the paper, such that it would probably be hard for the readers to reproduce e.g. equation (5.33). Again it would be helpful if the authors could be more precise.

In the last sentence of their conclusions the authors bring up the interesting question whether factorization is peculiar of the XXZ model and related models or rather a general property of integrable models. Here it could be mentioned that discrete functional equations of the type considered in the paper are know in a rather general context of quantum-group related vertex models,

[R6] Reduced qKZ equation: general case, A. Klümper, Kh. Nirov, A. Razumov, J. Phys. A 53: 015202, 2020

but an algorithm for connecting, say, the three-site functions with the two-site functions, as proposed by the authors, is unknown in this general case. In fact, I would expect that the equations proposed in [R6] do not uniquely fix the structure of the correlation functions.

Requested changes

See report

---

## Round 1 · Referee Report · Anonymous (Referee 3) · 2021-6-2

Strengths

First new and very interesting results on reduced density matrices of integrable face models and of their factorization properties

Weaknesses

On the presentation of the existing literature, on the notations and on the completeness of the steps in their proofs

Report

The manuscript starts the analysis of reduced density matrices of integrable face models with inhomogeneity parameters by deriving a discrete version of the quantum Knizhnik-Zamolodchikov type equation. The authors use this equation and some further unicity solution requirements to infer a factorized form for these correlation functions for (R)SOS models and a quantum chains of non-Abelian $su(2)_3$ Fibonacci anyons. This analysis produces explicit expressions of the reduced density matrices on two and three lattice sites.
The results are new and very interesting and they open the way to the generalization of the factorization properties of these correlation functions beyond those so far derived for the XXZ quantum spin chains. I recommend the publication of the paper in SciPost after the authors take cure of some improvements going from a more detailed presentation of the existing literature (in particular, on these correlation functions of related models) to a more detailed description of the used notations up to the clearness and completeness of the steps used in their proofs.
The two previous referees have done a very good job in identifying these improvements and their reports contain mainly the full list of required modifications and clarifications I have myself developed during the study of the manuscript. So, my advice is that the authors implement them. Let me just cite a couple of them that I would like to evidence. I agree with referee 1 on the fact that the large use of graphical notations (this also in the proofs) can result in a difficult access to a wide audience and the paper does not seem completely self-contained in the explication of them. I share exactly the same impression of referee 2 that some steps in the proof are missing to consider their statements/results completely proven. One is the argument of the analytic prolongation which implies the unicity of the solution, they have a proof of this properties or they are making some well educated guess/conjecture? This is essential to prove the factorized form of these correlation functions.
A part from that, I found somehow misleading, in the framework of integrability, the use of the terminology “generalized transfer matrices” for the reconstruction of local operators. The transfer matrices of integrable models in general define the family of commuting operators while this is not the properties of local operators. The reconstructions here presented are rather the generators of the dynamical algebra dressed by true transfer matrices. At page 10 the sentence “For more complicated cases this expression needs to modified” should contain a missing “be”.

Requested changes

As described above

---

## Round 2 · Referee Report · Anonymous (Referee 2) · 2021-7-27

Report

The authors have responded very positively to all three referee reports. In my understanding they have managed to considerably improve the quality of the presentation of their important results. The embedding of their achievements into the context of the existing literature is also much better now. I recommend the publication of the manuscript in its present form.

---

## Round 2 · Referee Report · Anonymous (Referee 3) · 2021-8-2

Report

The authors have implemented the main requirements and further improved the presentation of their results. The manuscript seems now suitable for publication on scipost.

---

## Round 2 · Referee Report · Anonymous (Referee 1) · 2021-8-15

Report

The main changes requested by the referees have been implemented, and the quality of the paper is now much better, both concerning references to the existing literature and clarity of the presentation and notations. I now recommend the publication.

---

## Round 2 · Author Response

We thank the referees for their positive feedback and the helpful suggestions.

---

## Round 2 · List of Changes

• we have rewritten and extended the introduction significantly to put our work into the proper context of existing literature, in particular on correlation functions for SOS models (Referee 1: References to previous works) and the factorization of correlation functions of the XXZ model (Referee 2).

Referee 1:

"Notations and definitions could sometimes be improved."

  • We have implemented the suggestions. To discriminate between indices representing states in the auxiliary space and corner values we have changed the notation for the former in (2.7)

  • explicit analytical formulae have been added to (2.12)-(2.16) and (4.1)

"About the solution of the inverse problem"

  • The solution of the inverse problem for CSOS models in Refs. [25,66] relies on the formulation of these models as a dynamical vertex model. In our proof of Theorem 1 the existence of a dynamical $R$-matrix is not used. The corresponding formulations in the introduction and Section 3 (p.7) are changed accordingly.

  • The proof of Theorem 1 for general L>2 is given in Appendix A.

"About the application of the functional equation"

  • We expect no problems with the solution of the functional equations for more general models (at least for finite systems). To study the factorization property in these cases, however, we would need a suitable ansatz which for the models considered in our paper is motivated by results from the XXZ model, i.e. (5.32). (no change)

  • Our results for the CSOS model are obtained for the reference state. Due to the simple nature of this state the 2-site correlations are given in terms of 1-point functions alone. We have added results on the 3-site density matrix where the same behaviour is observed.

Referee 2:

"Uniqueness of the solution to the discrete functional equations"

  • we have, in fact, observed (although this is not proven) the equivalent of the "asymptotic reduction" in certain topological sectors, see Eq. (5.30). As stated below Eqs (5.29) and (5.47), however, we find that it is sufficient to use the (weaker) relation between $D_N$ and $D_{N-1}$ obtained by taking partial traces for the recursive calculation of the density matrices in a given eigenstate of the transfer matrix (at least for the $r=4$ and $5$ RSOS models considered in the paper). We have clarified the corresponding statement in the conclusion.

  • Number of unknown functions in the 'physical part' of the correlation functions: For the $r=4$ and $r=5$ RSOS we find that two functions, in the topological sectors with quantum dimension a single function is sufficient. Preliminary results for models with $r>5$ indicate that the latter holds there, too. This is formulated as a conjecture in the conclusion.

  • We have added a comment on the characterization and calculation of the two-point function $f(\lambda,\mu)$ in the conclusion

  • Details on the algorithm for the computation of the structure coefficients have been added on p.18/19.

  • A Reference to [R6] has been added in the conclusion.

Referee 3:

  • On the terminology "generalized transfer matrices": We introduce this notation in the context of generic (not necessarily integrable) face models on p. 5,6. Here the concepts of a transfer matrix generating a family of commuting operators or of generators of the dynamical algebra do not exist. The difference between (2.8), (2,9) and (2.10) is only in the boundary conditions in the horizontal direction. That's why we have chosen this name (or "transfer matrices with generalized boundary conditions" in the introduction). (no change)

---

## Editorial Decision

published